



# The impact of far-reaching offshore cluster wakes on wind turbine fatigue loads

Arjun Anantharaman[1,2], Jörge Schneemann[1,2], Frauke Theuer[1,2], Laurent Beaudet[3], Valentin Bernard[3], Paul Deglaire[3], and Martin Kühn[1,2]

[1]Carl von Ossietzky Universität Oldenburg, School of Mathematics and Science, Institute of Physics
[2] ForWind - Center for Wind Energy Research, Küpkersweg 70, 26129 Oldenburg, Germany
[3]Siemens Gamesa Renewable Energy, Rouen, France

**Correspondence:** Arjun Anantharaman (arjun.anantharaman@uol.de)

**Abstract.** With the number of commissioned and planned wind farms increasing rapidly, analysing wind farm cluster wakes becomes essential for resource assessment and lifetime considerations. Cluster wakes influence wind turbine power in downstream wind farms in certain meteorological situations. Our objective is to ascertain whether far-reaching cluster wakes ($15\,\mathrm{km}$ to $21\,\mathrm{km}$) impact individual turbine loading in a downstream wind farm, considering the influence of atmospheric stratification.

We utilised SCADA data from an offshore wind farm and accelerometer measurements as the load proxy in the absence of load measurements to check short-term fatigue loading effects. We compared the absolute values of relevant SCADA variables of turbines in and out of the cluster wake. We found that while cluster wakes increase fluctuations of rotor speed and power, the load effects were lower than from turbines in the free-wind, primarily due to lower wind speeds. We developed a new methodology to quantify loads of turbines affected by the cluster wake while separating the dependency of loads on the inflow wind

speed. The turbines within the cluster wake showed a small increase in the load effects ($\approx 2.5\%$) when compared to turbines in free-wind, but lower than loads of turbines within the wind farm affected by inner-farm wakes (both at same local inflow wind speeds). We also found atmospheric stratification and the inflow wind speed to have no impact on the magnitude of loads within the cluster wake. Additionally, we found no additional blade mode excitations due to the presence of the cluster wake from the analysis of load spectra. We conclude that wind turbines affected by cluster wakes have a marginal increase in loads

when compared to reference conditions in undisturbed inflow. The absolute load effects in the cluster wake are lower due to the lower wind speeds. We propose the use of additional data from load sensors to further determine possible lifetime fatigue effects of cluster wakes on offshore wind turbines. These new insights can potentially add to the design standards of future wind farm clusters.

## 1 Introduction

In wind farms, single wind turbine wakes combine and form a wind farm or cluster wake downstream (Porté-Agel et al., 2020). Such cluster wakes can last several tens of kilometres and can influence downstream wind farms. While the impact on downstream wind farm power came in the focus of research in recent years, the influence on loads is rather unexplored. Both effects should be understood and modelled for better resource assessment, maintenance planning and lifetime considerations.



With the increase in offshore wind farm capacity and the commissioning of more wind farms to achieve climate targets (IEA,
2021), the wakes of entire wind farms or wind farm clusters (known as cluster wakes) are receiving more attention from the
wind energy industry and research. These cluster wakes have been measured through various sensing techniques, such as
Doppler Lidar (Schneemann et al., 2020), Doppler Radar (Nygaard and Christian Newcombe, 2018), Satellite SAR (Hasager
et al., 2015), and research aircrafts (Platis et al., 2018). These wakes have also been observed using wind farm power from
the Supervisory Control And Data Acquisition systems (SCADA), which are present in all operational wind farms. The wind
speed deficits due to these cluster wakes can be as high as 40 % of the free-stream velocity 20 km downstream of the cluster
from which it originated (Schneemann et al., 2020). Cluster wakes travel longer in stable atmospheric stratification, while in
unstable stratification, they dissipate faster due to higher ambient turbulence intensities (Platis et al., 2021). This points to the
importance of understanding the potential effect of large-scale wakes not only on power but also on the turbine loads, as the
disturbed flow downstream of a wind farm impacts several others.

SCADA data provides the unique opportunity to extensively assess these wake effects for every individual turbine in a wind
farm over a longer period, subverting the necessity for long in-situ measurement campaigns. It has been primarily used to
assess the power performance and wake effects of individual turbines (Gonzalez et al., 2019). Vera-Tudela and Kühn (2017)
used existing SCADA signals to evaluate lifetime fatigue and found that with an error lower than 1.5 %, a good prediction
of fatigue loads was possible for the blade-root flap-wise and edge-wise bending moments. They did, however, note that the
predictions were worse in waked conditions, though only marginally. The standard method of storing data from wind farms is
10-min averages, but the data from the sensors mounted on the turbines is usually sampled at higher frequencies which has been
used successfully in wind power and wind farm flow forecasting (Rott et al., 2020; Lin and Liu, 2020). Mittelmeier et al. (2017)
analysed turbine power from high-frequency SCADA data to estimate the atmospheric stability using the normalised power
fluctuations, similar to the well-known Turbulence Intensity (TI). High-frequency data is necessary to estimate the Damage
Equivalent Loads (DEL) and for spectral analysis, wherein the excited modes and fundamental frequencies can be analysed.
Noppe et al. (2016) used 1 s SCADA and accelerometer data to model the thrust loads and obtained a good match between
measured and predicted loads. Recent studies (Pettas et al., 2021) used meteorological data from the FINO1 met mast and
SCADA data to determine that the turbines in the Alpha Ventus wind farm experienced increased loading due to the operation
of neighbouring wind farms. The tower bottom load variations were found to be impacted by inter-farm interactions, with
the load effects decreasing as the distance from the other wind farms increased. They also found that the standard deviations
of the pitch angle and generator speed of the turbine showed increased values due to the wakes from the surrounding wind
farms (maximum distance of 6.5 km away), while the DEL at the tower base had no significant changes. As the study took
annual averages as a point of comparison, the atmospheric stability, which is a key driver of cluster wakes' persistence in the
atmospheric boundary layer, was not considered. According to IEC 61400-1 (2019) the possible load effects of neighbouring
wind turbines has to be taken into consideration by an increased turbulence intensity or dynamic wake model (no wind speed
reductions necessary) for upstream distances of up to 10 $D$, where $D$ is the upstream turbine rotor diameter.





Several studies have shown that atmospheric stability and increased turbulence intensity impact the fatigue loads on turbines due to single turbine wakes (Sathe et al., 2013; Churchfield et al., 2012). Holtslag et al. (2016) state that atmospheric stability has to be carefully considered in the analysis of fatigue loads since there are relatively higher fatigue effects for unstable

atmospheric stratification. Even though Pettas et al. (2021) analyse load effects at inter-farm distances of up to 7 km from the generating cluster, the cluster wakes have already been shown to persist as long as 55 km in weakly unstable stratification and even longer in stable atmospheric conditions (Schneemann et al., 2020). The first step in analysing potential cluster wake-induced load effects is to determine if there are short-term fatigue loads (10 min) which are adversely impacted. It is thus essential to investigate the effects of cluster wakes on wind turbine loads for far-reaching cluster wakes (>15 km) and determine

if (and how much) the atmospheric stratification affects turbine loading.

Our objective is to determine if far-reaching cluster wakes impact individual downstream wind turbine short-term fatigue loading dependent on the atmospheric stratification. We aim to address this objective by the following methodology:

- Firstly, classifying cluster wake situations by quantifying the caused power deficit based on SCADA data of a pair of wind farm clusters in the German North Sea

- secondly, assessing possible load impacts of the cluster wakes on individual turbines by developing a new methodology to compare load effects in the same wind speed conditions

- and finally, determining if the prevalent atmospheric stability can affect the magnitude of turbine load effects caused by cluster wakes

It is beyond the scope of the paper to investigate long-term lifetime fatigue effects which are heavily dependent on various wind

farm and site-specific influences. We use the standard deviation of several operational parameters and represent all the standard deviations of variables as follows: $u'$ is the standard deviation of the wind speed $u$, for ease of reading unless specifically mentioned otherwise. The paper is structured as follows: Section 2 introduces the reference wind farms, the wind farm and atmospheric data used in the analysis and highlights the steps involved in choosing a cluster wake case. Section 3 showcases a new methodology to specifically quantify the fatigue load effects caused by the cluster wake, which involves the selection of

a load proxy from SCADA and the metrics to analyze the effect of the cluster wake on wind turbine loads in all the analysed cases. Section 4 contains the results quantifying cluster wake effects on loads using SCADA data. Section 5 discusses the implications of the results and whether they could influence wind farm planning. Section 6 presents the conclusions of the study, with recommendations for further research.

## 2   Methodology

In this section, we describe the used data set from the cluster wake situation in between the N-6 and N-8 wind farm clusters in the German Bight. Additionally, we also detail how a cluster wake situation is selected and the steps involved in choosing a case, based not only on the wind farm parameters but also the atmospheric stability estimates.



## 2.1 Wind farms and operational data

An increase in the number of erected and planned offshore wind farms in areas such as the German Bight causes wind farms
to be situated close to each other, as in Fig. 1. We analyze the wake generated by the N-6 cluster hitting the N-8 cluster in the
German Bight area (see Fig. 1 and Table 1). The wake generated by the N-6 cluster for frequently occurring south-westerly
winds directly affects the inflow of the N-8 cluster, measured previously using lidar (Schneemann et al., 2020). A cluster
wake is a wake generated by a large number of wind turbines from one or more neighbouring wind farms (Platis et al., 2018;
Schneemann et al., 2020). We use operational SCADA data from the sister wind farms Albatros and Hohe See in the N-8
cluster in the period from 10-Jan-2021 till 17-Mar-2022 to analyze turbine loads affected by the N-6 cluster wake. Albatros
and Hohe See utilize the same wind turbines of type SWT 7.0-154. Both farms will together be referred to as A/HS in the
following. For the spectral analysis in Sect. 4.4 we used 10-Hz SCADA data of the turbines in the A/HS wind farm. For all
other analyses, we utilised 10-min statistics (mean and standard deviation) computed by the SCADA system.

Figure 2a shows the N-8 cluster and the region to the southwest down to the N-6 cluster. A wind direction sector from 230°
to 270° was chosen for the N-6 cluster wake analysis. This sector was specifically selected such that approximately half the
turbines in the front row of A/HS would be within the wake and the other half in free-wind conditions, to better understand
the difference in turbine behaviour due to the wake. The front row of A/HS is defined for this work as the 22 turbines which
are directly impacted by the N-6 cluster wake, highlighted blue in Fig. 2b. For the purpose of comparison, we chose turbines
in free-wind (green) and inner-farm wake (red) as reference conditions in the same 230° to 270° wind direction sector. The
wind direction was derived from the 10-min mean nacelle positions, and the resulting distributions for the nacelle positions
for normally operating turbines in A/HS across the 15-month analysis period is shown in Fig. 3. All nacelle positions were
corrected for any positional errors or offsets in values, e.g. due to calibration issues and the wind direction sector chosen for
cluster wake cases was also observed to be frequent.

**Table 1.** Wind farm specifications of the N-6 and N-8 clusters, detailing the names of the wind farms within each cluster, the turbine type, the rotor diameter $D$, the turbine's hub heights, the rated powers of both the turbine $P_{\text{turb}}$ and the entire wind farm $P_{\text{farm}}$ and the number of turbines in each wind farm.

| Cluster | Wind farm | Turbine | $D$ [m] | Hub height [m] | $P_{\text{turb}}$ [MW] | No. of turbines | $P_{\text{farm}}$ [MW] |
|---------|-----------|---------|---------|----------------|------------------------|-----------------|------------------------|
| N-6 | Bard Offshore I | BARD 5.0 | 122 | 90 | 5 | 80 | 400 |
| N-6 | Deutsche Bucht | MHI Vestas V164-8.4MW | 164 | 108 | 8.4 | 31 | 260 |
| N-6 | Veja Mate | SGRE SWT-6.0-154 | 154 | 103 | 6 | 67 | 402 |
| N-8 | Albatros (A) | SGRE SWT-7.0-154 | 154 | 105 | 7 | 16 | 112 |
| N-8 | Hohe See (HS) | SGRE SWT-7.0-154 | 154 | 105 | 7 | 71 | 497 |
| N-8 | Global Tech I (GT1) | Adwen AD 5-116 | 116 | 92 | 5 | 80 | 400 |





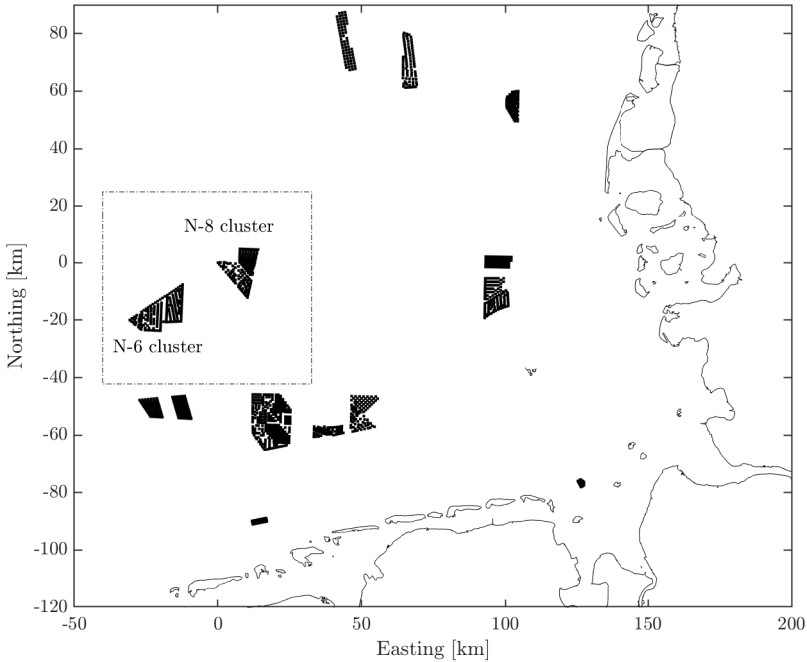

**Figure 1.** Map of operational wind farms in the German North Sea at the time of analysis [Jan 2021 - March 2022]. The N-6 and N-8 clusters are labelled and highlighted by a dashed box. The origin is defined as the turbine on the North-Western edge of the N-8 cluster.

## 2.2 Selection of cluster wake cases

We obtained cases of cluster wakes approaching A/HS by choosing an appropriate wind direction sector (230° to 270°) where at least eight A/HS turbines would be within the wake region, assuming a straight-line advection of the cluster wake velocity deficit. To limit our analysis to cases with turbines in normal operation we filtered out the data in situations of curtailment, turbine maintenance, shut down, and wind speeds below cut-in and above-rated.

Next, we considered situations only when the averaged nacelle position remained almost constant ($\pm 5°$) for at least 60 minutes for two reasons. Firstly, as we did not have measurements of the wind direction or the nacelle positions of the upstream N-6 cluster, we assumed that the cluster wake travels with approximately the same wind direction as estimated from the nacelle position of front-row turbines of the A/HS wind farm. The lowest wind speed considered for a cluster wake is 6 ms$^{-1}$, and we estimated the duration for the cluster wake to propagate between the wind farms as one hour (21 km/6 ms$^{-1} \approx 60$ min). Secondly, fixing the wind direction for at least one hour avoids situations with significant wind direction changes due to large-scale weather phenomena. Figure 4a displays a satellite SAR snapshot (ESA, 2021) of a suitable cluster wake situation, while Fig. 4b shows the time series of the nacelle position and wind speed for the same exemplary wake situation. The bounds for the wind direction where the cluster wake is relevant are shown with black dashed lines, and the shaded box shows the 80-min

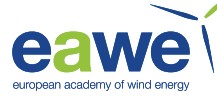
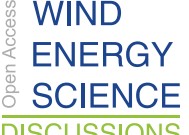


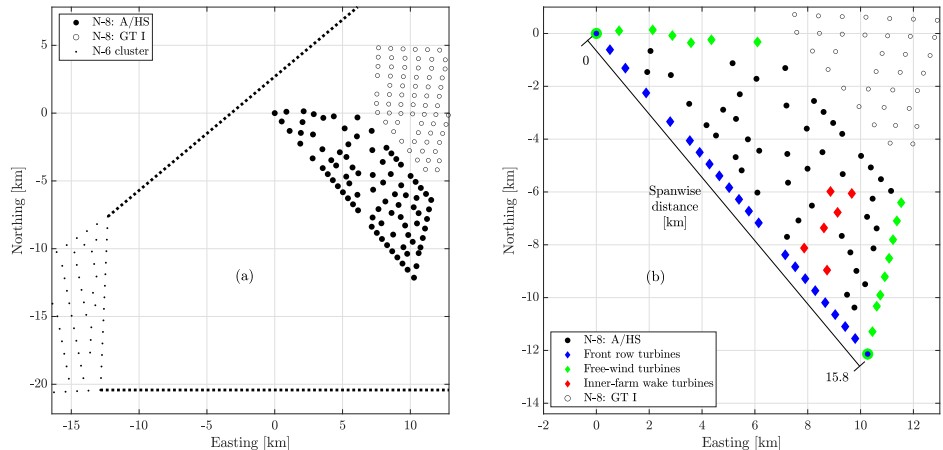

**Figure 2.** (a) Region of the N-6 cluster (dots) and the N-8 cluster (filled and empty circles). The wind direction sector considered for N-6 cluster wake analysis spanning from 230° to 270° is indicated by dashed lines. (b) A/HS wind farm in the N-8 cluster with highlighted turbines for analysis: front-row turbines to the southwest (blue diamonds), reference turbines in free inflow (green diamonds) and turbines in inner-farm wake conditions (red diamonds). The origin of the coordinate system is the front-row turbine in the northwestern corner. The span-wise distance along the front row of turbines is also indicated, starting from the north east corner at [0,0] until the turbine at the southern corner, 15.8 km away from it.

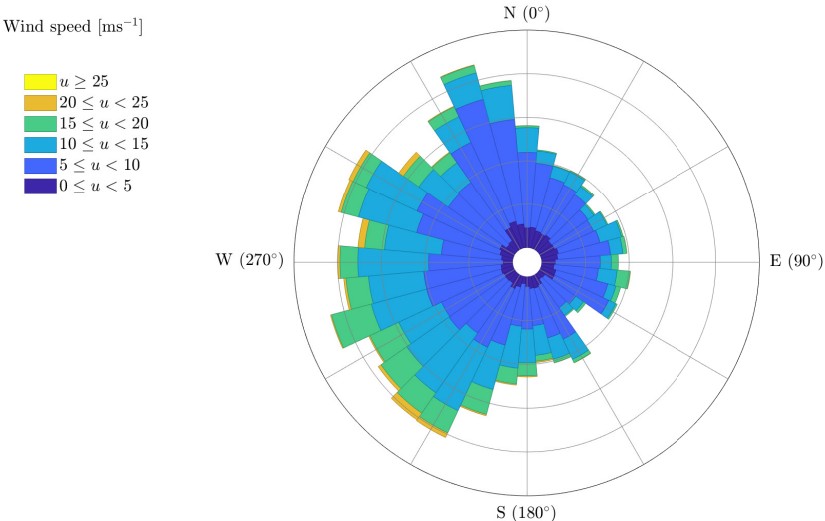

**Figure 3.** Polar plot showing the 10-min mean nacelle positions of all the turbines in A/HS from 10-01-2021 till 17-03-2022. The plot differs slightly from the ten year mean wind rose (not shown here) due to the period of analysis.

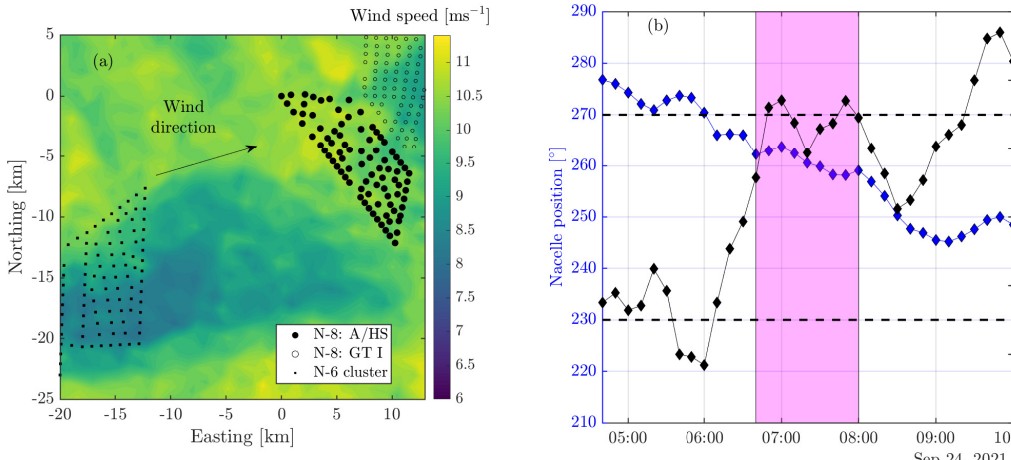

**Figure 4.** (a) SAR snapshot of the N-6 cluster wake impacting the N-8 cluster on 24 Sep 2021, 05:57:43 (ESA, 2021). (b) Time series of the averaged nacelle position (left y-axis) and the corresponding wind speed (right y-axis) of the front-row turbines for the same situation in (a). A suitable time period for a cluster wake case is highlighted in pink, with a nearly constant wind direction. The dashed lines represent the 230° to 270° wind direction sector considered for cluster wake scenarios.

period (06:40 to 08:00) where the nacelle position (ergo wind direction) is suitable for further analysis. The wind speed in the front row of turbines which are not affected by the cluster wake are also within $0.5 \ \mathrm{ms}^{-1}$ of each other. There are four cluster wake cases (out of 96) where this does not hold true since the free-wind turbines are operating above the rated wind speed. We did not discard these cases since the turbines within the cluster wake were still below-rated and satisfied all other conditions.

After selecting periods with the desired wind direction, we apply a rigid set of filters to ensure that each chosen case corresponds to a cluster wake with normal wind farm operation. The different steps are outlined in Fig. 5. We only had access to the Veja Mate wind farm SCADA data from the N-6 cluster, so cases were only classified as cluster wake when Veja Mate was producing power. Further, we checked the difference in power of the front row of the N-8 cluster (A/HS). Only cases where the magnitude of the power deficit caused by the cluster wake in the front row, $\Delta P$, was higher than 0.5 MW (7% of the rated turbine power) were considered for further analysis. We manually checked each selected case to verify that the power across the front row had the expected range of values, and if at least half (11 out of 22 turbines) of the front-row turbines were producing power. We ended up with 96 cases of interest where at least 17 out of 22 turbines were in normal operation, though the more conservative criteria of at least 11 (half of the total) operating turbines for the consideration of a case was surpassed (see Fig. 5).





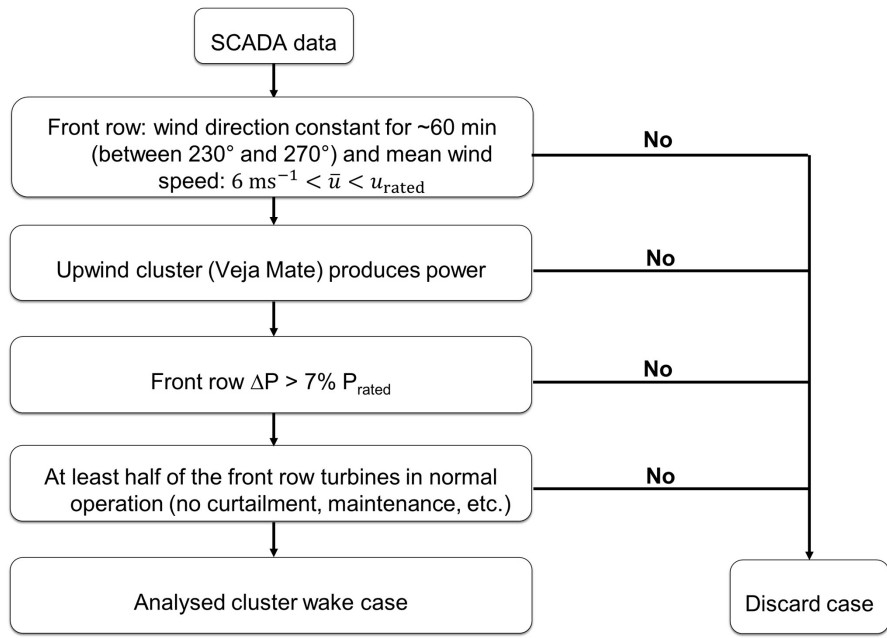

**Figure 5.** Flowchart depicting the series of steps used to select a case to be considered "cluster wake" for load analysis.

### 2.3 Atmospheric stability estimates

The recovery of wakes is highly dependent on the atmospheric stability. Cluster wakes in particular have a stability dependence, persisting longer in the Atmospheric Boundary Layer (ABL) for stable stratification and dissipating faster in unstable

conditions. There are several methods to estimate the atmospheric stability in field campaigns, and it is particularly difficult to get comprehensive measurement data in front of wind farms without met-masts. We follow the classification of stability regimes as in Sathe et al. (2013), but with reduced number of regimes to simplify the analysis. Table 2 shows the used regimes along with the number of analyzed cluster wake cases.

We use the Monin Obukhov length $L$ to estimate atmospheric stability from simulations with the Weather Research and

145 Forecasting model (WRF, version 4.2.1) (Skamarock et al., 2019) on a grid point between the N-6 and N-8 clusters. Simulations used the Fitch wind farm parameterisation (Fitch et al., 2012) and included all operating wind farms in the German Bight. We compared WRF stability values against atmospheric measurements, where the atmospheric stability was estimated using temperature differences between the sea surface and 25 m above the sea surface at the Global Tech I wind farm in the N-8





cluster (Schneemann et al., 2020) and found a good match. Since such atmospheric measurements were unavailable for the
period of the current study, the WRF simulation data (Cañadillas et al., 2022) was used.

**Table 2.** Stability regimes' classification and the number of analysed cluster wake cases.

| Stability | Monin Obukhov length range [m] | Cluster wake cases |
|---|---|---|
| Stable | $500 > L > 0$ | 29 |
| Near neutral | $|L| > 500$ | 13 |
| Unstable | $-500 < L < 0$ | 54 |

## 3 Quantifying cluster wake-caused loads: a new methodology

For cluster wake situations, quantifying the potential fatigue load effects by directly comparing turbine loads within the cluster
wake to turbine loads in undisturbed inflow is not possible due to the different wind speeds at the turbines. Reference loads are
thus required to remove the dependency on wind speed and atmospheric conditions (turbulence intensity, atmospheric stability).
This makes a new approach necessary to differentiate the load effects in cluster wake situations to those from turbines which
are in free inflow situations or affected by inner-farm wakes. Wind turbine fatigue loads are typically compared by load spectra
or Damage Equivalent Loads (DEL) derived from rain-flow counting of load time series. A comparison with the design loads
according to IEC 61400-1 (2019) or between different site and operational conditions is done for short-term (10 min) fatigue
loads or long-term fatigue loads extrapolated from a set of load situations with individual frequency of occurrence to the design
lifetime. We limit the comparison to the analysis of short-term load fatigue loads in different atmospheric conditions. In this
section we first introduce the load variable utilised from the available SCADA data, then detail the methodology to compare
and quantify cluster wake-caused turbine loading to reference loads in non-cluster wake situations. Finally, we also introduce
a performance parameter to conveniently represent the magnitude of the cluster wake-caused loads.

### 3.1 Selection of load proxy in SCADA

Quantitative highly resolved data is needed from all the wind turbines in the farm to analyze the potential load effects of cluster
wakes. This is only possible through SCADA. As the turbines in A/HS were not equipped with load sensors, we turned to the
available SCADA measurements that could act as a proxy for fatigue loads. Environmental and wind turbine variables which
are sensitive to fatigue loads are the inflow turbulence intensity and the standard deviations of the nacelle accelerations $a$,
pitch angle $\beta$, rotor speed $\Omega_{\mathrm{rotor}}$ and turbine power $P$ (Cosack, 2010; Mittelmeier et al., 2017; Pettas et al., 2021). Nacelle
accelerations are measured in the fore-aft (fa) and side-side (ss) directions and their standard deviations are written as $a'_{\mathrm{fa}}$ and
$a'_{\mathrm{ss}}$ respectively in this manuscript. When comparing these potential load proxies against the mean wind speed, only the nacelle
accelerations had both a reasonably low and uniform spread of values and an approximately linear relation to the below-rated
wind speed. Additionally, we conducted an independence study to confirm if the load proxies mentioned above were consistent
regardless of the choice of turbine and found that only $a'_{\mathrm{fa}}$ was sufficiently independent of turbine choice in the A/HS wind





farm. One explanation could be the fact that since the sensors are located on the nacelle, the primary driver of loads is in the the fore-aft direction due to thrust and so is dominant. Therefore, we proceeded to use only the fore-aft nacelle accelerations as a proxy for wind turbine fatigue loading.

Table 3 shows the correlations of the load proxy $a'_{\mathrm{fa}}$ against calculated fatigue quantities such as the damage equivalent flap-wise blade root bending moment and the tilt and yaw moments of the tower top from both load measurements ($R_{\mathrm{meas}}$) and 180 simulations ($R_{\mathrm{BHawC}}$). The $R_{\mathrm{meas}}$ were derived from the DEL calculated from strain gauge measurements of the turbines in the Borssele wind farm, also located in the North Sea. Although the turbines in the Borssele wind farm have a different rated power, they have the same rotor diameter and the comparison only serves to confirm the choice of the load proxy where load measurements are available. $a'_{\mathrm{fa}}$ is very well correlated to the DEL of the flap-wise blade root bending moment and the yaw and tilt moments of the tower top. Additionally, we also used simulation data from the highly resolved Siemens Gamesa 185 in-house aero-elastic model BHawC (Rubak and Petersen, 2005; Muller et al., 2023) for the same turbine type as in the A/HS farm to compute correlations ($R_{\mathrm{BHawC}}$) between the $a'_{\mathrm{fa}}$ and DEL variables. The measurements and simulations show good agreement, in that $a'_{\mathrm{fa}}$ is well correlated to DEL quantities of the blade and tower top. This is also consistent with the findings of Cosack (2010), wherein the nacelle accelerations' fluctuations and other SCADA signals were correlated to the magnitude of wind turbine DELs via neural networks. Further studies (Vera-Tudela and Kühn, 2017; Pérez-Campuzano and Gallego-190 Castillo, 2018) also support the choice of the standard deviation of the nacelle accelerations as a good indicator for the DEL experienced by the turbine.

**Table 3.** Pearson's correlation coefficient $R$ between the load proxy $a'_{\mathrm{fa}}$ and the Damage Equivalent Loads (DEL) for the blade and tower, calculated from measurements from the Borssele offshore wind farm in North Sea and BHawC simulations for the A/HS wind farm. The DEL was calculated from rain flow counting of load measurements on the turbines.

| Load variables | $R_{\mathrm{meas}}$ - Borssele [-] | $R_{\mathrm{BHawC}}$ - A/HS [-] |
|---|---|---|
| $a'_{\mathrm{fa}}$, blade flap-wise moment | 0.85 | 0.80 |
| $a'_{\mathrm{fa}}$, tower top tilt moment | 0.87 | 0.51 |
| $a'_{\mathrm{fa}}$, tower top yaw moment | 0.85 | 0.79 |

### 3.2 Reference wind turbine loads in non-cluster wake scenarios

In the partial load range, fatigue loads are approximately proportional to the wind speed and turbulence intensity (Pettas et al., 2021). Cluster wakes are defined as situations with significant wind speed reductions which might be associated with increased 195 wake turbulence. Hence, any assessment of cluster wake-induced fatigue loads should try to separate these two effects. When using the selected load proxy $a'_{\mathrm{fa}}$ to quantify fatigue load effects due to the cluster wake, it is essential to have a reference $a'_{\mathrm{fa}}$ free from the influence of the cluster or inner-farm wakes. For this purpose, we created a look-up table of Standard Loads (SL) for the $a'_{\mathrm{fa}}$ in both free-wind (no wake effects, referred to as SL-free) and inner-farm wakes (when a turbine is directly influenced by one or more upstream turbines within a farm, referred to as SL-inner) situations. For both scenarios, appropriate



wind direction sectors were chosen, with sixteen turbines used for free-wind and six turbines for inner-farm wake cases, shown in Fig. 2b with green and blue markers, respectively. The SL tables were created from more than 2.5 years of SCADA data, from 01-Jan-2020 till 31-Jul-2022. The wind direction sectors chosen were 60° to 80°, 100° to 120° and 300° to 360° due to the absence of any offshore wind farms upstream in a range of 80 km (see Fig. 1). In the utilized sectors, the nearest coastline is more than 100 km away, and so we assume no coastal influence. There could be larger-scale phenomena arising from the

land-sea transition (Schulz-Stellenfleth et al., 2022), but these are neglected in the current study. The SL-inner table was created by considering the six turbines within the N-8 cluster (see Fig. 2b), for the same wind direction sector as SL-free.

Figure 6 shows the correlation between the $a'_{\mathrm{fa}}$ of the selected SL-free and SL-inner turbines with the other measured variables in SCADA data ($\beta$ is the turbine pitch and $\Omega_{\mathrm{rotor}}$ is the rotor speed). The mean wind speeds in the whole analysis refer to the 10-min measurements from the anemometers on the turbine nacelle which includes a correction function to account for the

placement of the nacelle anemometer behind the rotor. We found that the turbulence intensity (TI) calculated from the nacelle anemometer wind speeds was not well correlated to $a'_{\mathrm{fa}}$, one reason possibly being the positioning of the anemometer behind the rotor. This means that while $a'_{\mathrm{fa}}$ is correlated to $u'$, it is poorly correlated to the nacelle anemometer-based TI. We also included a variable called "POwer normalized TI" (POTI), defined by Mittelmeier et al. (2017) as the standard deviation of the turbine power normalised by the mean turbine power for 10-minute averages. This was found to be an indicator of atmospheric

stability from SCADA parameters, which also is a key driver of cluster wakes. Additionally, POTI was found to also be a better indicator of turbulence than the nacelle anemometer-based TI (Barthelmie et al., 2007). However, both the nacelle anemometer TI and POTI were not found to be well correlated with $a'_{\mathrm{fa}}$. In the end, we considered only two parameters to construct the SL table: $u$, to account for the wind speed range of the turbine operation and $u'$, which was consistently a highly correlated parameter with $a'_{\mathrm{fa}}$ across both scenarios of free-wind and inner-farm wakes. We also found a similar correlation of the power

fluctuations $P'$ as $u'$, but we did not utilise that for the SL tables since the spread of values was inconsistent across wind speeds. This could be attributed to the high sensitivity of the power fluctuations' correlations to the flow conditions (Seifert et al., 2021).

A two-dimensional SL table was created by binning $a'_{\mathrm{fa}}$ in 1 ms$^{-1}$ steps of $u$ and 0.1 ms$^{-1}$ steps of $u'$. Figure 7a shows the SL tables for $a'_{\mathrm{fa}}$ in free-wind conditions. Figure 7b shows the Standard Error of the Mean, SEM $= \sigma(a'_{\mathrm{fa}})/\sqrt{N}$, where $\sigma(a'_{\mathrm{fa}})$

is the standard deviation of $a'_{\mathrm{fa}}$ and $N$ is the number of samples in a particular bin of values. This gives an idea of how the uncertainties are distributed and most of the data is located where the SEM is low. Although the anemometers behind the rotor are subject to more fluctuations, there is enough data in each bin so that the SEM is low for most of the main wind speed values of interest. Any bin with lower than 100 values was discarded to avoid biasing the SL table with low-confidence bins. The SL-inner matrix (not shown) has much higher $u'$ values, due to inner-farm wake effects but the SEM distribution is similar to

SL-free.





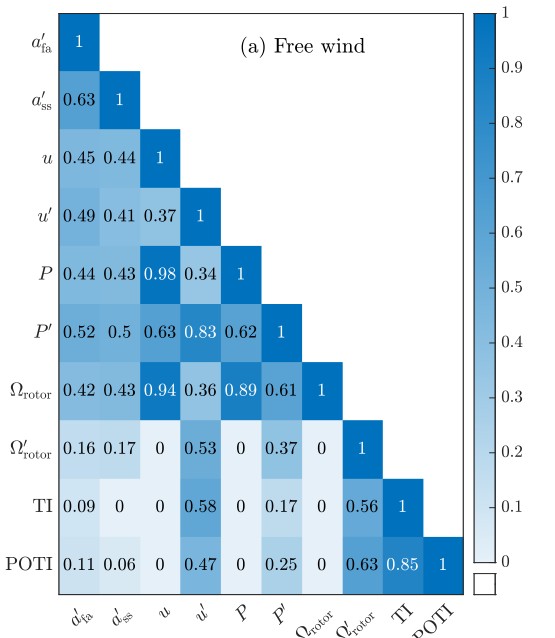

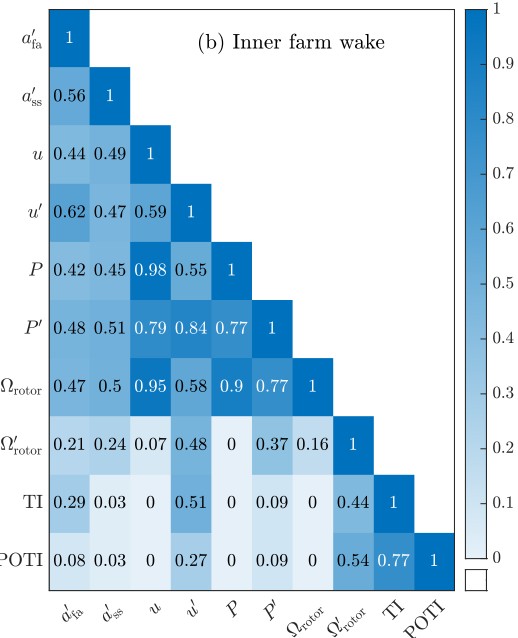

**Figure 6.** Pearson correlation coefficients $R$ of the measured and derived SCADA quantities to the load proxy $a'_{\mathrm{fa}}$ in free-wind (a) and inner-farm wake (b) conditions for below rated wind speeds. The Standard Load (SL) tables were built for $a'_{\mathrm{fa}}$ as a function of $u$ and $u'$ due to their $R$ values in both (a) and (b).

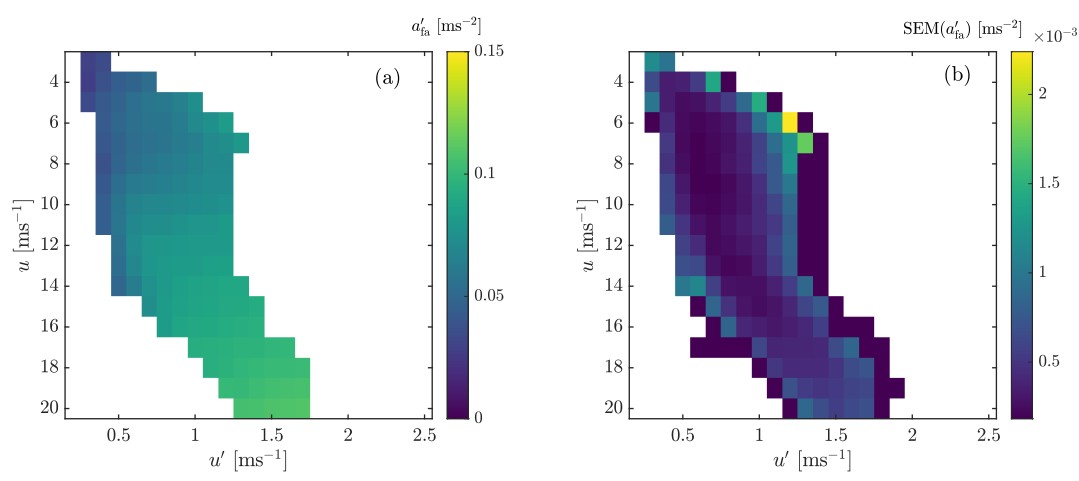

**Figure 7.** a) Standard Loads (SL) matrix for $a'_{\mathrm{fa}}$ in free-wind and b) the corresponding Standard Error of the Mean (SEM) both dependant on $u$ and $u'$.

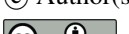



### 3.3 Performance indicators to quantify cluster wake-caused load effects

We created SL tables in Sect. 3.2 to compare load effects independent of the wind speed reduction in the wake. We defined two performance indicators, $\zeta_{\text{free}}$ and $\zeta_{\text{inner}}$ to represent how much the $a'_{\text{fa}}$ deviates from the SL-free and SL-inner standard load cases, respectively. These parameters indicate how much the $a'_{\text{fa}}$ for a turbine in the cluster wake quantitatively differs to the $a'_{\text{fa}}$ for the same $u$ and $u'$ values in the SL tables, making it easier to interpret the quantitative effects of the cluster wake-caused fatigue loading.

$$\zeta_{\text{free}}(u, u') = \left( \frac{a'_{\text{fa}}(u, u')}{a'_{\text{fa, SL}_{\text{free}}}(u, u')} - 1 \right) \cdot 100 \ [\%] \tag{1}$$

$$\zeta_{\text{inner}}(u, u') = \left( \frac{a'_{\text{fa}}(u, u')}{a'_{\text{fa, SL}_{\text{inner}}}(u, u')} - 1 \right) \cdot 100 \ [\%] \tag{2}$$

## 4 Results

The results are presented in three subsections: firstly, we show an exemplary cluster wake situation and the subsequent analysis using absolute turbine parameters and the Standard Load tables. Secondly, we also compare nacelle acceleration spectra to determine if any structural modes are excited at turbines in the cluster wake. Finally, we show the broader results for all the cases and analyse the role of atmospheric stability.

### 4.1 Exemplary wake case

We show the results from one exemplary cluster wake case for the effect on the turbine load proxy (24th of September 2021, from 6:40 till 08:00), shown in Fig. 4. The wind direction is 261°, and the average wind speed across all the wind turbines in the front row is 12.3 ms$^{-1}$. The atmospheric stability during that period was very unstable, with $L = -282$ m. The satellite Synthetic Aperture Radar (SAR) image for the same day (ESA, 2021) at 05:57:43 is shown in Fig. 4a. Even though the SAR image is a snapshot earlier than the period considered, it still serves to confirm the existence of the N-6 cluster wake.

Figure 8a shows the wind speed, power, $a'_{\text{fa}}$ and corresponding standard deviation of wind speed in the front row of turbines in A/HS due to the N-6 cluster wake influencing the farm. All three variables are normalized with their respective average of the three maximum values in the front row. Wind speed and turbine power are expected to be maximum when the turbines are not affected by wakes, as it is the case for the first half of the front row (span-wise distances from 0 km till about 8 km, first 10 of 22 front-row turbines). The more the cluster wake impacts the turbines, the reduction in wind speed and power is also accompanied by a reduction in the $a'_{\text{fa}}$. The $u'$ has some fluctuations, but is marginally increasing when the other three variables reduce due to the presence of the cluster wake, around 5 km span wise distance onward. Figure 8b compares the $a'_{\text{fa}}$ across the front row (blue line and markers, measurements) in the cluster wake case to the reference situations of free-wind





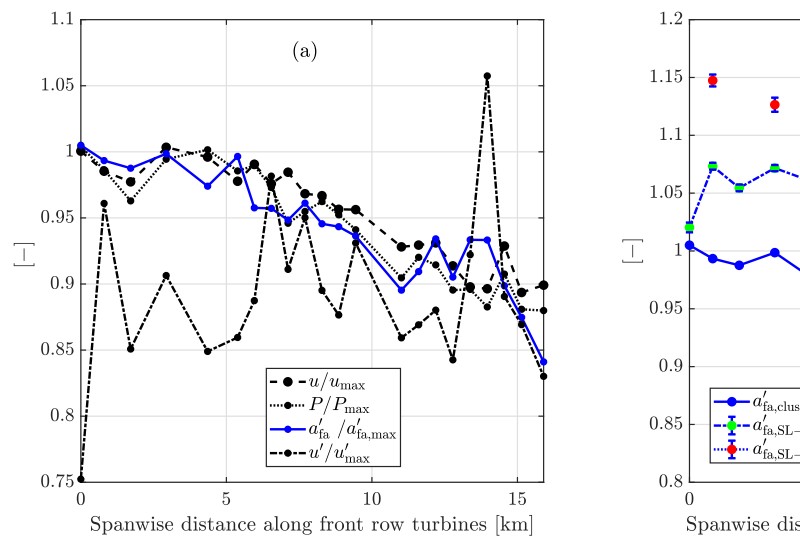

**Figure 8.** (a) Variation of $u$, $P$, $a'_{\text{fa}}$ (blue line) and $u'$ across the front row for the exemplary case as a function of the span-wise distance. Each variable is normalized by the average of the three maximum values in the front row. (b) Comparison of $a'_{\text{fa}}$ in the exemplary cluster wake case (blue line, same as in (a)) to the $a'_{\text{fa}}$ in reference situations of free-wind (SL-free, blue line and green markers) and inner-farm wake conditions (SL-inner, blue line and red markers) for the same local $u$ and $u'$. Error bars indicate the SEM of $a'_{\text{fa}}$ (for the SL tables only) and all the variables are normalized by the average of three maximum values in the front row of $a'_{\text{fa}}$ in the cluster wake case.

(green markers and blue line, SL table) and inner-farm wakes (red markers and blue line, SL table), per turbine for the same
$u$ and $u'$. Each $a'_{\text{fa}}$ is normalised by the average of the three maximum values of $a'_{\text{fa}}$ in the cluster wake case, such that the
blue dotted lines in both figures are the same for $a'_{\text{fa}}$. The $a'_{\text{fa}}$ for the exemplary wake case is overall much lower and there is
a reduction of $a'_{\text{fa}}$ for the turbines in the cluster wake (8 km to 15 km span-wise distance). Some missing values in the values
from SL-inner are due to the fact that not enough measurements were present in that bin of $u$ and $u'$ ($N < 100$, see Sec. 3.2).

The influence of the cluster wake is present in the front row beyond approximately 8 km span-wise distance (see Fig. 8a),
as observed from the reduction in turbine power. The accelerations from the SL tables remain approximately constant despite
wind speed reductions, while the cluster wake case does not follow the same trend. One potential reason for the overestimation
of $a'_{\text{fa}}$ by the SL tables could be due to the highly unstable atmospheric stratification and could point to more data required in
specific stability conditions. For this case study, the cluster wake does not have any adverse load effects and even has lower
$a'_{\text{fa}}$ than the free-wind turbines (it is not constant along the row like the SL tables), possibly due to the combined effects of
increased wake recovery in very unstable atmospheric stratification and lower wind speeds due to the cluster wake.

Finally, we computed the power spectrum of the nacelle accelerations, using 10 Hz SCADA data. This was done by computing
the power spectrum of the raw 10 Hz time series and then converting the units of the power spectrum to decibels. The power

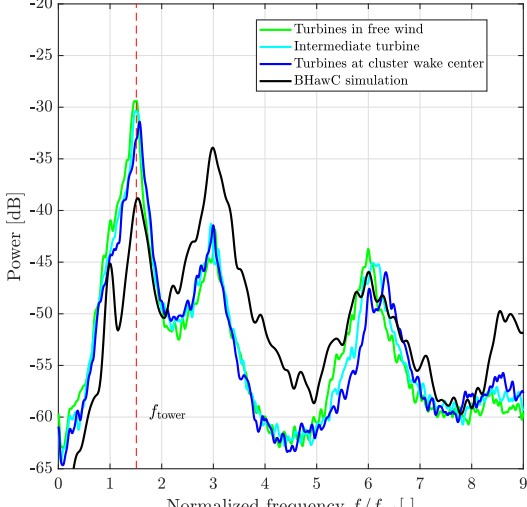

**Figure 9.** Power spectrum ($a_{\mathrm{fa}}$) of three turbines in the front row for the exemplary cluster wake situation: free-wind (green), turbine at an intermediate location between maximum and minimum power deficit (cyan) and the turbine at the cluster wake center (blue) plotted against the normalised frequency (with the rotational frequency) with the corresponding aero-elastic simulation from BHawC for the same wind speed of the turbine at the cluster wake centre. The first tower mode is marked with a red dashed vertical line.

spectra of the nacelle accelerations were compared to observe if any modes showed additional excitations by the cluster wake or if any other non-standard behaviour was present. Figure 9 shows the power spectrum over the normalized frequency for three

turbines of the front row. To simplify the analysis, we considered only three representative front-row turbines for analysing this cluster wake case: one turbine in free-wind with maximum power (green), one turbine within the cluster wake with minimum power (blue) and one turbine in the transition region between cluster wake and free-wind (cyan). In all the spectra, the peak with the highest amplitude is the eigen frequency of the tower ($f_{\mathrm{tower}}$), as seen in the peak at $f/f_{\mathrm{rot}} \approx 1.5$ (red dashed line in Figure 9) and its subsequent multiples. This case study does not show additional excitations for specific modes by the

cluster wake, as all the spectra overlap, also due to all the turbines having the same rotor speed. Additionally, we also show the spectrum obtained from the standard DLC 1.2 (normal turbine operation) aero-elastic simulation in BHawC for the same wind speed (black). The wind speed of the simulation spectrum is the same as the wind speed of the turbine in the cluster wake centre and the nacelle TI is also matched to be the same, at 6.5 %. Despite a small mismatch in amplitudes, the peaks of the spectrum occur at the same frequencies.





## 4.2 Cluster wake effects on absolute turbine variables

Each cluster wake situation is classified by the magnitude of the maximum power deficit ($\Delta P$) it causes, defined by the difference between the maximum and minimum turbine power along the front row. Figure 10a defines four sets of turbines (colours correspond to the same turbines in subsequent figures) to determine potential load effects caused by cluster wakes:

- free-wind turbines (**green**): turbines in free stream, and these include all turbines whose power ranges from the maximum in the front row, till 10% of the maximum power deficit is reached, $P_\mathrm{max} - 0.1 \cdot \Delta P \leq P \leq P_\mathrm{max}$.

- Cluster wake turbines (**blue**): turbines directly affected by the cluster wake and centred around the turbine in the front row with the lowest power, including those turbines that continue to experience at least 50% of the maximum power deficit, $P_\mathrm{min} \leq P \leq P_\mathrm{min} + 0.5 \cdot \Delta P$.

- inner-farm wake turbines (**red**): to get an estimate of the effect of inner-farm wakes, the same turbines that were used to create the SL-inner table are considered for the analysis of cluster wakes.

- last-row turbines (**magenta**): these were chosen to further analyse inner-farm effects as they contain the superposition of wakes of all turbine rows of the A/HS wind farm.

Figure 10b shows the absolute $a'_\mathrm{fa}$ values normalised with the mean $a'_\mathrm{fa}$ of the free-wind turbines across all 96 cases in the form of a box plot. The box plot displays the statistics across all the analysed cluster wake cases: the box edges represent the upper and lower quartiles, the line in the box middle is the median, the whiskers represent the minima and maxima that are not outliers and the outliers are the circular markers. We clearly see that the inner-farm and last-row turbines experience higher accelerations, while the turbines in the cluster wake are subject to marginally lower nacelle accelerations than the free-wind turbines. The SL tables (Fig. 7a) show lower $a'_\mathrm{fa}$ if the wind speed is reduced, which is occurring at all the turbines affected by the cluster wake.

Figure 11 shows the variation of the 10-min mean wind speed $u$ and normalised wind speed fluctuations $u'/u$ for all the cases. Since we use wind speeds from the nacelle anemometers, we do not refer to $u'/u$ as the TI, but rather as the nacelle TI. It is evident that the inner-farm and last-row turbines experience both lower wind speeds and simultaneously much higher nacelle TI. Turbines within the cluster wake also have much lower wind wind speeds, but slightly higher nacelle TI. Table 4 shows the summary of the average values of the four turbine types for all the variables of interest. The values represent the deviation on average to the free-wind turbines, so we observe that the cluster wake turbines have 15.5% lower wind speeds but 16.3% higher nacelle TI as compared for the same cases of free-wind turbines. There is a 16.3% increase in nacelle TI for the cluster wake affected turbines, as compared to 80 to 90% increase in the nacelle TI for the inner-farm and last-row turbines. This marginal increase could be due to an increase in the atmospheric TI due to the cluster wake, but could also be attributed to the uncertainties in using nacelle anemometers as a reference metric, which we discuss further in Section 5. We also show the effects on the last row of turbines, which are impacted by the aggregation of all upstream turbine wakes, and the qualitative results match the inner-farm turbines, which would experience only the wakes of a few turbines. We also made a



comparison of the turbines in the last row for SSW wind directions and found no increase in $a'_{\text{fa}}$, when splitting the last row into halves: one containing only inner-farm wakes and one containing the additional cluster wake caused velocity deficits. Though the turbine layout affects the propagation of inner-farm wakes, the similar effects on the inner-farm and last-row turbines in Table 4 indicate no combined effects of the cluster wake on inner-farm turbines.

In general, the values of both fore-aft and side-side (not shown) accelerations are lower for cluster wake-affected turbines, but all the other variables show an increase. On the one hand, the wind speed is lower in the cluster wake, but the wind speed fluctuations are higher, causing potentially higher fatigue loads. These two phenomena counterbalance each other, but we have to analyse this further, as absolute value increases are not significant if there is no reference for comparison and the magnitude of load effects are also much smaller than for the turbines inside the wind farm. The power and the rotor speed fluctuations are also uncertain parameters to draw conclusions as we found their performance to not be consistent as a function of wind speed and chosen turbine types. There could be biases in the data or other weather phenomena that could also affect the values for the free-wind turbines, leading to much lower fluctuations.

**Table 4.** Effects of the cluster wake on the three turbine types, expressed as percentage differences to the free-wind turbines. The values were obtained by grouping the turbines types and averaging over all the 10-min data.

| Parameter | Cluster wake turbines [%] | inner-farm turbines [%] | last-row turbines [%] |
|---|---|---|---|
| $u$ | -15.55 | -21.80 | -23.81 |
| $u'/u$ | 16.31 | 81.91 | 90.71 |
| $a'_{\text{fa}}$ | -7.14 | 21.28 | 21.27 |
| $a'_{\text{ss}}$ | -1.51 | 1.47 | 2.04 |
| $P'$ | 12.70 | 59.58 | 68.72 |
| $\Omega'_{\text{rotor}}$ | 50.23 | 189.47 | 231.42 |

## 4.3 Cluster wake effects quantified using reference loads

In the previous section we found that, on average, the cluster wakes do not increase the nacelle accelerations, and even cause a reduction in loads, due to the lower wind speeds within the wake region. We created reference load tables in Sect. 3.2 to compare load effects independent of the wind speed and defined two performance indicators $\zeta_{\text{free}}$ and $\zeta_{\text{inner}}$ in Sect. 3.3 to conveniently quantify fatigue load effects caused by the cluster wake using the SL tables in free-wind and inner-farm wake situations.

Figure 12 displays the averages of $\zeta_{\text{free}}$ and $\zeta_{\text{inner}}$ as a function of $u$, $u'/u$ and $\Delta P/P_{\text{rated}}$ in each bin along with the standard deviations of each bin as error bars. We do not show the last-row turbines in Fig. 12 to 14 as their load effects are similar to the inner-farm turbines. The distributions of both $\zeta_{\text{free}}$ and $\zeta_{\text{inner}}$ for cluster wake turbines (blue) significantly differ from inner-farm turbines and are quite close to those of the free-wind turbines. There is also no dependency of the cluster wake turbine loads on $u$ and $u'$. This can be observed from the near-zero lines for the free-wind and cluster wake-affected turbines


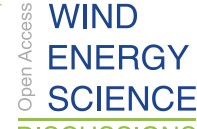


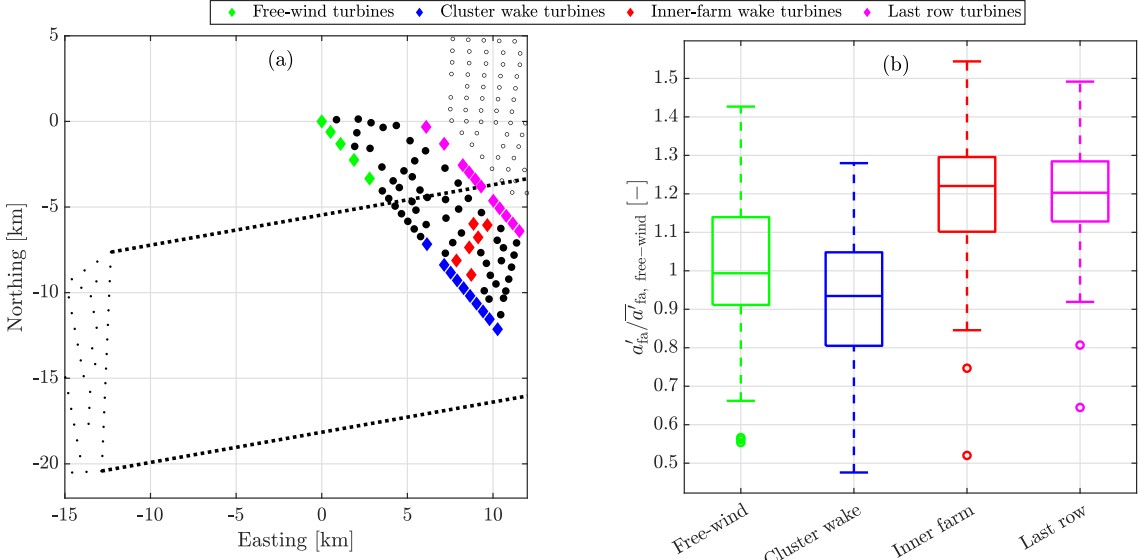

**Figure 10.** (a) Top view on N-6 (black dots) and A/HS wind farm for 260° wind direction. In A/HS, the free turbines (green), cluster wake turbines (blue), inner-farm turbines (red) and last-row turbines (magenta) are marked as turbine types for analysing load effects. (b) Absolute $a'_{\mathrm{fa}}$ values normalised with the mean of free-wind turbines for all four turbine types (same colors as in (a)) across all the analysed cases, showing a significant increase in $a'_{\mathrm{fa}}$ for turbines inside the wind farm and the last row, both experiencing the effects of inner-farm wakes. The scatter points outside the whiskers of the box plot are outliers.

in Fig. 12[a, b, d, e]. Figure 12c and Fig. 12f show the performance indicators $\zeta$ as a function of the normalised cluster wake power deficit. An increasing magnitude of the (normalised) cluster wake power deficit seems to marginally cause higher loads. It is to be noted that the power deficit in these two plots differs for each turbine, as opposed to one uniform power deficit for a one-hour cluster wake situation which was shown in Sect. 4.1. This is evident from the free turbine lines (green) which do not proceed beyond zero, as they are the turbines that do not see a power deficit in the front row. Qualitatively, there is no effect

of the cluster wake on loads, dependent on these three parameters. To quantify the effect, we calculated the mean of the entire distribution of points and the results are summarised in Table 5. The uncertainty in the SL tables is also evident from these results, as $\zeta_{\mathrm{free}} = -0.79\,\%$ for free-wind turbines and $\zeta_{\mathrm{inner}} = 5.25\%$ for inner-farm turbines, which should be zero in the ideal case. Despite these biases, the cluster wake turbines are significantly lower in load effects than the inner-farm wake turbines. In the second row of Table 5 the values are corrected by these biases. Considering this correction, the cluster wake turbines

have 2.4% higher loads when considering $\zeta_{\mathrm{free}}$ (second table row, correcting for the bias), not accounting for any atmospheric effects. The $\zeta_{\mathrm{inner}}$ shows almost similar values for turbines in the cluster wake and free-wind turbines.



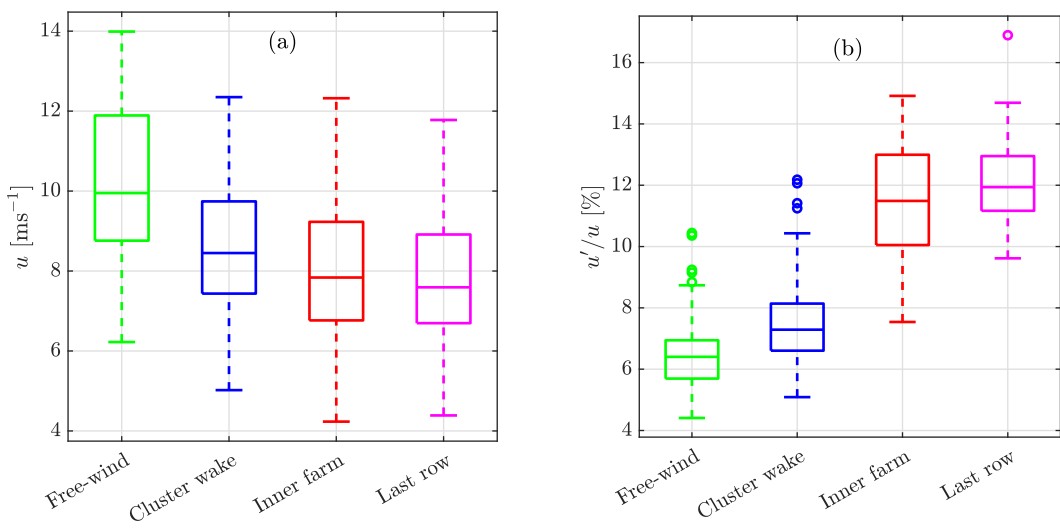

**Figure 11.** (a) Distribution of absolute wind speed $u$ and (b) normalised wind speed fluctuations $u'/u$ for all cases. The wake affected turbines (blue, red and magenta) have lower $u$ and higher $u'/u$ as compared to the free-wind turbines (green). The colour coding follows Fig. 10.

**Table 5.** Cluster wake effects on $a'_{\mathrm{fa}}$, expressed in deviations from the SL tables.

|  | $\zeta_{\mathrm{free}}$ [%] | | | $\zeta_{\mathrm{inner}}$ [%] | | |
| --- | --- | --- | --- | --- | --- | --- |
|  | Free-wind | Cluster wake | Inner-farm | Free-wind | Cluster wake | Inner-farm |
| $a'_{\mathrm{fa}}$ biased | -0.79 | 1.61 | 15.64 | -1.85 | -2.18 | 5.25 |
| $a'_{\mathrm{fa}}$ corrected | 0 | 2.40 | 16.43 | -7.10 | -7.43 | 0 |

Atmospheric stability is an important parameter that affects the wake recovery of single turbine wakes and also cluster wakes. We compared the performance parameters as a function of atmospheric stability, shown by the bar graph in Fig. 13. The uncertainties are highest for unstable stratification, shown by the highest biases, potentially due to higher fluctuations in the atmosphere. The deviation from the SL tables for inner-farm turbines is still much higher than the other two turbine sets, but the differences between the free turbines and cluster wake turbines are once again marginal. In all the cases, wind turbines affected by the cluster wake experience slightly higher loads ($\approx 2.5\%$) than the turbines in free-wind conditions when the lower wind speeds in the cluster wake are accounted for using performance indicators.

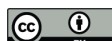
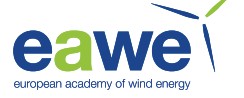


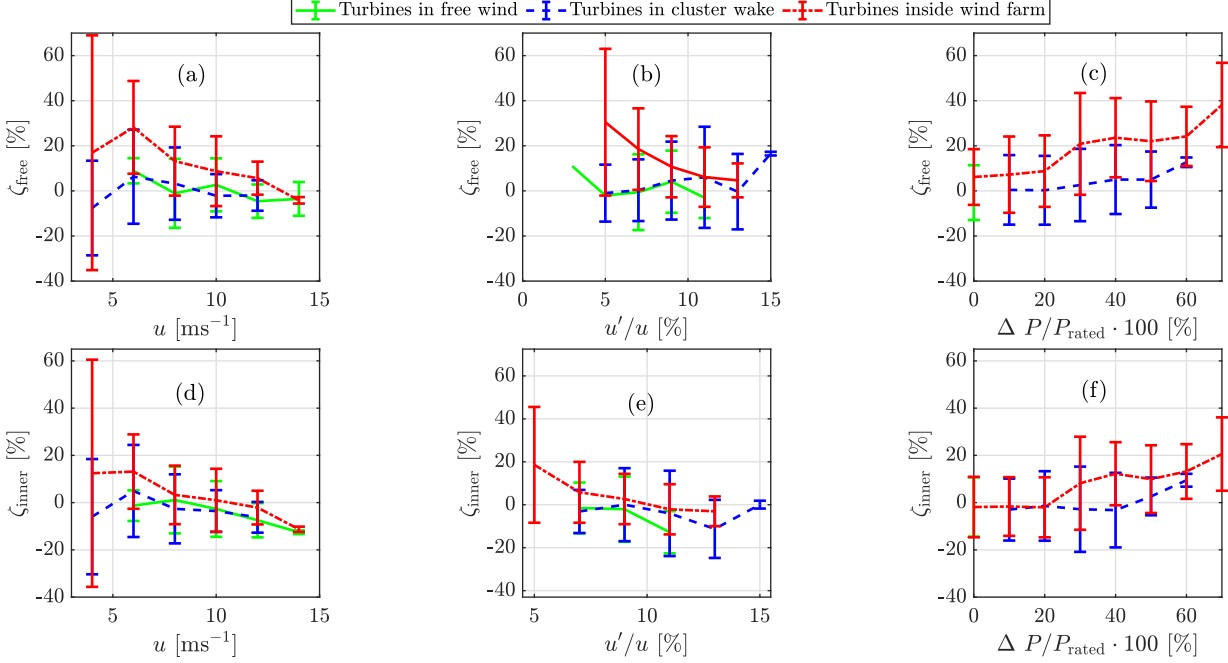

**Figure 12.** Binned averages of the performance indicators $\zeta_{\text{free}}$ and $\zeta_{\text{inner}}$ as a function of $u$ (a and d), normalised fluctuations of the wind speed $u'/u$ (b and e) and the magnitude of a turbine's power difference to free stream turbine power, expressed in a percentage of rated power (c and f). The error bars are the standard deviations of $\zeta_{\text{free}}$ and $\zeta_{\text{inner}}$ for each bin.

### 4.4 Spectral analysis

We used the 10 Hz nacelle fore-aft acceleration data to obtain the power spectrum for the different turbine types with the aim to quantify the dynamic response of the turbines to the cluster wake. We considered the same three turbines as in the exemplary cluster wake situation (see Sect. 4.1): two turbines with maximum and minimum power respectively, and one lying in the intermediate section, all in the front row. Since cluster wakes do not usually have a well-defined boundary, the intermediate turbine is taken as a representation of the transition between the free stream and maximum cluster wake-caused velocity deficit.

We then averaged the spectra for these turbines across all the cases, and the result is shown in Fig. 14a. Since the spectra for the different turbines across all cases are at different rotor speeds, the spectra are plotted against the frequency normalised by the first tower eigen frequency. All the spectra overlap, and there are no modes excited at turbines either inside the cluster wake or close to the wake boundary. This is across all the wind speeds and atmospheric conditions, so excited frequencies in certain conditions could be averaged out. We thus quantified the ten highest peaks in each spectrum. The histograms of the excited

frequencies are shown in Fig. 14b. The amplitudes of the usual mode peaks are not significantly changed across the turbines affected by the cluster wake (cyan and blue spectra in Fig. 14).





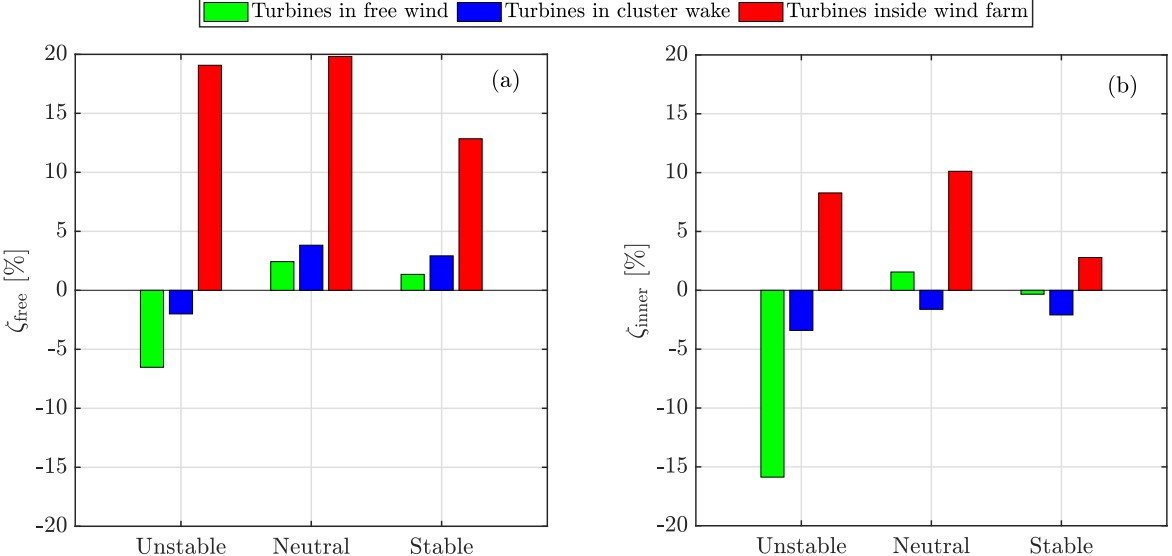

**Figure 13.** Bar graphs showing the performance indicators averaged over all turbines in a set for different atmospheric stability regimes, (a) $\zeta_{\text{free}}$ and (b) $\zeta_{\text{inner}}$.

This is, however, insufficient to conclude on frequency excitation as there could be single situations with additional mode excitations. We manually went through all the cases where the spectra were available (56 out of the 96 cases) and determined only two situations where the spectra differed. The first was when the rotor speed differed significantly between the three turbine types, causing shifted peaks at the harmonics. The second situation when the spectra did not match was close to cut-in, when several factors could affect turbine performance (such as pitching). Nevertheless, only three cluster wake cases out of 56 presented with mismatched spectra and we did not analyse these in further detail as the cluster wake turbine did not present with additional frequency excitations.

## 5   Discussion

We used SCADA data to determine if cluster wakes affect a short-term fatigue load proxy of offshore wind turbine response and also classified the effect based on atmospheric conditions. The load proxy was found to be lower in cluster wake-affected turbines due to the lower wind speed, but marginally higher when compared to reference situations of turbines in free-wind conditions. We also analysed the nacelle acceleration spectra and found no increased response amplitudes at certain frequencies at the cluster wake affected turbines below-rated wind speeds. We discuss the implications of these findings, along with the validity of using said load proxies and the limitations of the current analysis.





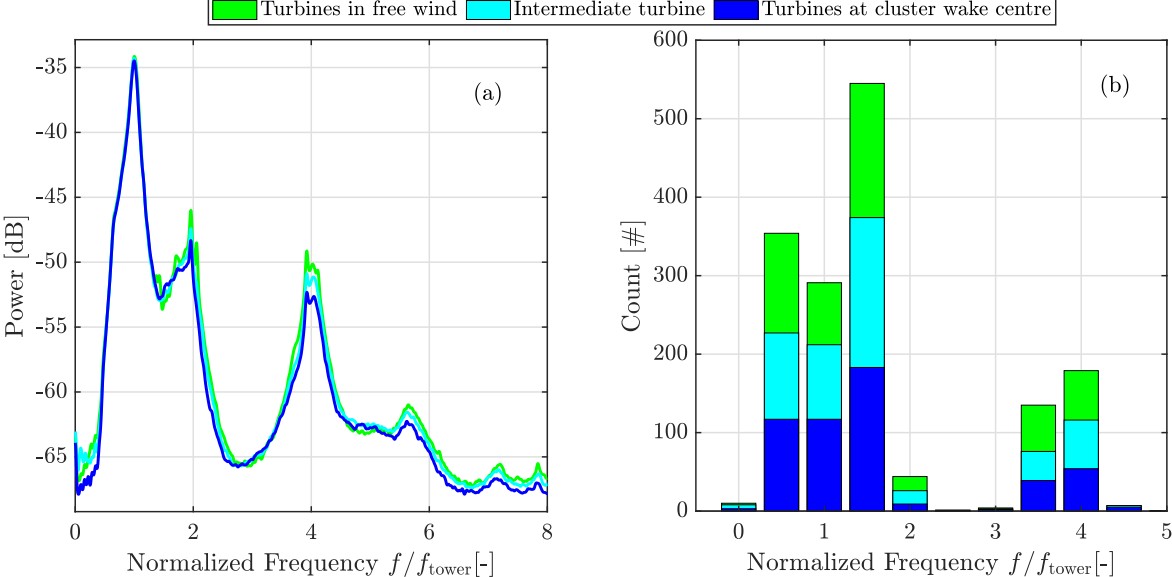

**Figure 14.** (a) Averaged spectra of the three turbine types as in Fig. 9. The spectra across all the cases are averaged and peaks overlap at the same frequencies, normalised with the first tower fundamental frequency. (b) Histogram showing the distribution of the ten highest amplitude peaks excited for all the cases and turbine types, again showing no additional excited modes due to the cluster wake. The bar-histogram plots are stacked on top of each other to represent all the peaks of each turbine type.

Pettas et al. (2021) found the maximum distance that the wakes from neighbouring wind farms impacted SCADA signals was 6.5 km. They found that the DEL did not show any significant changes at these distances, and the only parameters that showed increased values due to the cluster wake were the pitch and the generator speed. Firstly, we found a small increase in the nacelle TI for the turbines in the cluster wake more than 15 km downstream, but the increase could also be attributed to the location 390 of the nacelle anemometers. Secondly, we found the generator speed fluctuations increased due to the cluster wake and this could be due to the increased fluctuations in the wind speed. Our load proxy ($a'_{\mathrm{fa}}$) shows a reduction in absolute values. The lower wind speeds due to the cluster wake seem to be the primary contributors to load reduction overshadowing the effect of the increased fluctuations in wind speed. This is, however different when the reductions in the wind speed are taken into account and the loads are compared to a reference at local wind conditions. For this purpose, we introduced two performance 395 parameters $\zeta_{\mathrm{free}}$ and $\zeta_{\mathrm{inner}}$. We found a 2.4 % increase in these parameters caused by the cluster wake. We further confirm that despite very large power differences (up to 50 % of rated power) caused by the cluster wake, the load proxy is not impacted, even when compared to reference turbines in free-wind conditions. There is a marginal increase in the performance parameters used to analyse loads with an increase in the power deficit, but Fig. 12c and Fig. 12f also have much higher deviations in values, as seen by the wide error-bars. In all the cases considered, the cluster wake-affected turbines seem to be marginally 400 affected by the cluster wake, though not as significant as the inner-farm wake-affected turbines when reduction in wind speeds





are accounted for within the cluster wake region. The marginal load increase could be attributed to the increased fluctuations of the wind speed within the wake, but further analysis from load sensors and more cluster wake situations are needed to quantify these load effects.

### 5.1 Nacelle accelerations as a load proxy and comparison of spectra

Nacelle accelerations have been shown to be a very good proxy for loads since they are significantly impacted by the turbulence intensity (Cosack, 2010). We also performed correlation studies with load measurements from two offshore wind farms and determined that the fore-aft nacelle accelerations were an appropriate choice to analyse load effects due to cluster wakes. We compared nacelle acceleration spectra from aero-elastic simulations with BHawC to spectra from load signals and found that the nacelle acceleration spectra are most similar to the tower base and tower top fore-aft bending moment spectra. However,

these are simulations for onshore turbines and do not take into consideration hydrodynamics or any potential yaw misalignments which could affect the spectrum. Despite the minor offset in spectral power between simulations and measurements, the peaks in the simulated spectrum are the same when compared to high-frequency nacelle accelerations from SCADA. When there are rotor speed differences, the spectral peaks consist of the fundamental tower mode, along with harmonics of the blade passing frequencies, which are clearly distinguishable, except close to cut-in wind speeds. The difference in the wind speeds

due to the cluster wake does not impact the shape of the spectra or excited frequencies, and possible regions where an effect could occur are closer to cut-in wind speeds. This is also where pitching affects turbine operation and is not taken into account in either the generation of the SL tables or in the analysis of the spectra. There could also be startup effects of the turbine, so analysing this wind speed region becomes complex and requires a large amount of data. Most of the cases we analysed in load spectra were in unstable atmospheric stratification (10 Hz data available for 56 out of 96 cases), so we were unable to

distinguish the spectral peaks between different atmospheric stratifications.

### 5.2 Data limitations and scope for further research

The reference loads (SL tables) were created from more than 2.5 years of SCADA, classified in bins of $u$ and $u'$. Each bin was only considered valid when there was a balance between a sufficient number of data points and a low standard error of the $a'_{\mathrm{fa}}$ in the same bin (see Sect. 3.2 ). In spite of these conditions, Fig. 13 shows that biases are present in the calculated performance

indicators. This is approximately 6 % (unstable conditions) for free-wind turbine's $\zeta_{\mathrm{free}}$ in Figure. 13a and 9 % (unstable and neutral conditions) for inner-farm turbine's $\zeta_{\mathrm{inner}}$ in Figure. 13b, both of which should be zero, since they contain the reference loads used to create the SL tables. One reason for this could be the higher ambient turbulence due to increased mixing in the atmosphere, leading to less correlated $a'_{\mathrm{fa}}$ behaviour as a function of $u'$. Another reason for increased biases could be the binning of the nacelle accelerations as a function of $u$ and $u'$, both measured by nacelle anemometers. St Martin et al. (2017)

found that transfer functions are required to be corrected for the nacelle anemometer-measured statistics. These corrections were found to be higher for unstable atmospheric stratification and higher turbulence intensities, which is also where the biases in the SL table comparisons are higher. $\zeta_{\mathrm{inner}}$ also shows increased bias, which also points to it not being a reliable indicator of load effects due to cluster wakes. This could mean creating SL tables either requires more parameters as inputs when there is



higher turbulence due to ambient or single wake effects or that even more data is required for reference conditions of inner-farm
wakes.

Table 4 shows that if only $a'_{\mathrm{fa}}$ is considered as a load proxy, then the load effects for the cluster wake situations are even lower
than those for the turbines in free wind. This could also be similar to a case study from Neumann and Emeis (2020) who found
reduced Turbulent Kinetic Energy (TKE) above a wind farm within the cluster wake region, even lower than the ambient TKE,
but more sensor data is needed to confirm the TKE effects for the analysed cluster. However, we have seen that when the values
are compared to a reference at the same local wind speeds, there are slightly higher loads. The results can be interpreted in two
ways: on absolute values, the cluster wakes have no increased load effects and can even be beneficial by causing lower loads,
due to lower wind speeds. However, when local conditions of wind speed are accounted for, there are slightly increased loads.
Determining whether this would affect lifetime fatigue loads requires more data, preferably from direct load measurements
representative for the complete set of site-specific environmental and operating conditions. We also noticed that the absolute
fluctuation values of the rotor speed and the power are increased, which is similar to the findings of Pettas et al. (2021), even
though their comparisons were long-term averages and wind speed binned. We found that the turbines within the cluster wake
on average had higher nacelle-based TI, though the location of the anemometers makes it difficult to conclude on the effects of
the cluster wake on inflow TI. Nacelle anemometer measurements have been shown to capture wake effects inside wind farms
(Kang and Won, 2015) and also the ambient TI, though with disturbance due to the effect of the rotor (Göçmen and Giebel,
2016). We found the resulting increased nacelle-based TI to be smaller for cluster wake turbines than for inner-farm turbines.
Another factor that could affect the inner-farm turbine values is the superposition of two effects: cluster wakes and inner-farm
wakes. There could be smaller effects in the row of turbines directly behind the front row, but it is difficult to distinguish an
already small load increase due to cluster wakes with inner-farm wake added to it.

Although we utilised a suitable proxy for loads, ultimately, strain gauge measurements on turbines in cluster wake situations
could further strengthen the conclusions. Ziegler et al. (2017) used only tower bottom strain gauge measurements and found it
sufficient for structural monitoring and lifetime assessment of offshore wind turbines on monopiles. It is also possible that the
cluster wake partially affects only one load parameter that was not directly correlated to the nacelle accelerations. Shaler et al.
(2023) state that the most important parameters for fatigue and ultimate load analysis of turbines are the ambient turbulence
and the vertical wind shear in the most dominant wind direction. Measurements of the TI at multiple locations would be ideal,
as it will answer the question of how much the cluster wakes impact TI, also in the dependency of atmospheric stability.
Furthermore, direct load measurements would enable DEL comparisons and long-term statistics can then be compared against
the simulated design loads. This will be particularly beneficial for the operators of wind farms that experience frequent cluster
wakes and turbine manufacturers to know the potential effects of such conditions on the design load envelope.

## 6    Conclusions

We aimed to experimentally quantify the effects of 15 km to 21 km long cluster wakes on the short-term fatigue loading
of offshore wind turbines dependent on atmospheric stability. Although cluster wakes cause wind speed deficits that lead to



significant power losses, we found that they only marginally affect the fluctuations of the nacelle fore-aft acceleration, which was used as a proxy for fore-aft turbine loading. In absolute values, turbines in a cluster wake experience lower load effects than turbines in free-wind conditions when considering the load proxy, due to lower wind speeds in the wake region. However, cluster-waked turbines show marginally higher load effects ($\approx$2.5%) as compared to free-wind turbines when comparing situations with the same local wind speeds and wind speed fluctuations. No significant dependency was observed of the loads on atmospheric stratification, mean wind speed, fluctuations in wind speed or the magnitude of the cluster wake deficit, though the uncertainties are higher in unstable stratification due to limited data. Spectral analysis of high-frequency nacelle acceleration data showed no increase in the frequency peaks in the spectrum due to cluster wakes, while the overall mean spectral shapes also showed negligible differences.

The current analysis shows only marginal effects of cluster wake-induced short-term fatigue loading on wind turbines with the available load proxy of fore-aft nacelle acceleration fluctuations at same local wind conditions. While the conclusion that the cluster wakes do not excite modes is beneficial from a design point of view, there are also limitations to which the data set is subject. Damage Equivalent Loads (DEL) calculations were not performed for short-term or long-term statistics since the analysed wind farm turbines did not have strain gauges for load measurements, but DEL is an important metric for fatigue analysis. Further work can also be directed at intermediate distances between wind farms (9 km to 15 km), along with even more cases of confirmed cluster wake occurrence. Future research should use actual load measurements and determine if the increased fatigue effects warrant an amendment of the design standards for load analysis and wind farm site assessment.

*Author contributions.* AA, JS, LB, VB, PD and MK conceived the idea of the research. AA developed the methodology, performed the data analysis and wrote the manuscript. MK supervised the work. JS, LB, VB, FT and MK contributed with several fruitful discussions. All co-authors thoroughly reviewed the manuscript.

*Competing interests.* The contact author has declared that neither they nor their co-authors have any competing interests.

*Acknowledgements.* This research has been partly supported by the Federal Ministry for Economic Affairs and Climate Action in the framework of the research projects X-Wakes (FKZ 03EE3008D) and C2-Wakes (FKZ 03EE3087B) on the basis of a decision by the German Bundestag and by the European Union's Horizon 2020 research and innovation program under the Marie Sklodowska-Curie grant agreement No. 858358 (LIKE – Lidar Knowledge Europe, H2020-MSCA-ITN-2019). We would like to thank Martin Dörenkämper for providing WRF data and Bastien Duboc for simulation data and discussions regarding BHawC.





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
