# Peer review of "The impact of far-reaching offshore cluster wakes on wind turbine fatigue loads"

_Wind Energy Science, 2025_

## Referee Comment (RC2)

**Summary :**

The paper entitled "The impact of far-reaching offshore cluster wakes on wind turbine fatigue loads" investigates the impact in terms of loads of a cluster wake on wind turbines located from 15 to 21 km downstream. The authors utilize SCADA data and turbine-based measurements and derive a load proxy based on the available measurements to investigate the fatigue loads.

They found that the standard deviation of nacelle acceleration correlated well with certain DELs (blade flap-wise moment, tower top moments), and so used this proxy to investigate the loads for turbines affected by the cluster wake.

For this research work, they developed a methodology for quantifying loads on a turbine as a function of the incoming wind speed: while turbines in cluster wakes show a small decrease of loads compared to free wind turbines, separating the dependence of loads on the incoming wind speed leads to a small increase of fatigue loads for cluster wake turbines.

The also found that atmospheric stratification has no impact on the magnitude of loads within the cluster wake. There were also no additional blade mode excitations due to the presence of the cluster wake.

They conclude that wind turbines affected by cluster wakes have a marginal increase in loads when compared to turbines in freestream conditions (reference freestream conditions).

**General comment**

The paper is well written and pleasant to read. The research work is very interesting, as it investigates farm interactions based on measurements, which is always valuable for the wind community.

A large part of the paper is dedicated to the methodology and how to quantify cluster wakecaused loads: how "cluster wake" cases are determined from all available data, how the proxy is determined, how the stratification is estimated etc. It is Interesting and important because this helps the reader to understand the complexity of SCADA data processing and the limitations of the available data (compared to numerical investigations, where all the flow field is known). These two sections are dense, with a lot of information, and some explanations/details are sometimes missing (see Specific comments).

The analysis of the results is rigorous, and the discussion is critical, highlighting the limitations of the study. A comparison with numerical studies would be interesting, if such studies exist in the literature.

Please see below for more specific comments

**Specific comments:**

1. Page 3, lines 65-66, Introduction : "Our objective is to determine if far-reaching cluster wakes impact individual downstream wind turbine short -term fatigue loading dependent on the atmospheric stratification."

I would adapt/modify this sentence as the study of dependance on atmospheric stratification is only a small part of the analysis.

2. Page 4, line 94, Section 2.1 : "... data from the sister wind farms Albatros and Hohe See...".

The authors should also refer to Fig 2 to help to the reader locate wind farms of interest in the N8 cluster.

- Page 4, lines 103-104, Section 2.1 : "we chose turbines in free-wind (green) and innerfarm wake (red) as reference conditions..." At this stage of the reading, the meaning "reference conditions" is not clear. This will become clearer later, but for the sake of readability, this should be explained a little more here.
- 4. Page 4, line 105, Section 2.1 : "The wind direction was derived from the 10-min mean nacelle positions".

Which turbines were used to calculate the wind direction? All turbines?

5. Page 5, lines 114-115, Section 2.2 : the authors wrote that they filtered out situations where wind speed was higher that the rated speed. What is a situation with above rated wind speeds: is it when all upstream turbines are impacted by an above-rated wind speed? The authors should clarify this, especially as later in the article, they write that if turbines in free wind speed are in above rated conditions, that is not a problem (see comment 8)

And why do the authors discard situations where the wind speed is higher than the rated speed?

- 6. Page 7, Figure 4 (a): I guess the authors do not have a snapshot a little later, to really see the wake of the cluster that will impact the downstream wind farms. The snapshot should be explained a bit more, to be useful in the analysis/paper. Do we see in figure 4 (a) a part of the cluster wake that will have an impact on cluster N8?
- 7. Page 7, Figure 4 (b): Are these wind speed and direction for each wind turbine?
- Page 7, lines 124 to 127 : "The wind speed in the front row of turbines which are not affected by the cluster wake are also within 0.5 m/s of each other".
   Is it a condition for the selection of a cluster wake case? And I do not understand the

sentence that follows (same paragraph):

"There are four cluster wake cases (out of 96) where this does not hold true since the free-wind turbine are operating above the rated wind speed."

What does the term "this" refer to? Why does operating at a wind speed higher than the rated speed necessarily mean that the wind speeds of the free-wind turbines are not within 0.5 m/s of each other?

And final sentence:

"We did not discard these cases since the turbines within the cluster wake were still below-rated and satisfied all other conditions". Why do the authors keep these cases? (related to comment 5)

- 9. Page 7, lines 131-132 : the authors mention the power deficit caused by the cluster wake in the first line, but they do not detail how they calculate it. Is it calculated in the same way as the power deficit discussed in section 4.2? The definition of power deficit should be detailed here, and perhaps recalled later in the results.
- 10. Page 8, Section 2.3, second paragraph: If I understood correctly, the authors validated WRF simulations based on atmospheric measurements for another period, and they found a good match. So they use another WRF simulations for the considered period. Are there the results of Canadillas et al. ? I yes, the authors should write that it comes from this reference explicitly and maybe detail the reference in one sentence.
- 11. Page 9, lines 157-160, Section 3: This sentence is difficult to understand. Is this what you need to do for a complete fatigue analysis?

- 12. Page 9, lines 171-173, Section 3.1 : it is written that the nacelle accelerations have an approximately linear relationship with wind speed below the nominal value. What happens for wind speeds higher than the rated speed? (Because the authors wrote that they kept situations where the free wind turbines are higher than the rated speed).
- 13. Page 10, line 181 : for the Borssele wind farm, what period is used to compute the DELs? Do the authors compute correlations over a wide range of atmospheric conditions and stratifications? I assume that correlations are better for certain wind conditions... It would be useful to have more details about these calculations. Same comment for the numerical simulation. Further in the paper (Section 5.1, discussion), it is written that this is a numerical simulation for onshore turbine. It should be mentioned here.
- 14. For the DEL, do the authors correct the number of cycles based on the mean loads?
- 15. Page 10, Table 3 : How do the authors explain the difference in correlations between measurements and numerical simulations (for the tower top tilt moment)?
- 16. Page 11 : I do not understand the difference between u' and the anemometer-based TI. How are they computed respectively? The anemometer-based TI is computed based on u' no? Why is the correlation between a'fa and u' good, but not between a'fa and TI?
- 17. Page 12, Figure 7 : It would be interesting to add SL for inner farm effects even if it is not explained in detail. (but this would increase the size of the paper)
- 18. Page 14, lines 270 : "..., possibly due to the combined effects of increased wake recovery in very unstable stratification and lower wind speeds due to the cluster wake".
  I do not understand this sentence: the effects are opposite. Why would a combination of these effects lead to a decrease in loads?
- 19. Page 21, line 373 : the authors write that, for 56 out of 96 cases, spectra were available. What does this mean? Because the authors wrote on page 20 that they averaged the turbine spectra for all cases: so, "all cases" means 56 cases or 96?
- 20. Page 22, line 386 : "Pettas et al. (2021) found the maximum distance that the wakes from neighbouring wind farms impacted SCADA signals was 6.5 km". Whatever the size of the upstream wind farm? Or for one layout/situation?
- 21. Page 22, lines 388-389 : "... that showed increased values due to the cluster wake were the pitch and the generator speed. Firstly, we found a small increase in the nacelle TI..." I do not understand the link, and I think that "Firstly" is not the right term to begin this sentence. Do the authors mean that Pettas et al. found nothing at 6.5 km although they found impacts at 15 km ? It should be clearer. And next lines (lines 389-390) : the nacelle anemometer is located at the same position, whatever the turbine (in freewind or in cluster wake). So why might the increase of TI for cluster wake turbines be attributed to the location of the nacelle anemometer?
- 22. Page 24, discussion: are there any numerical studies of cluster wakes, which might help the authors to explain some of their results/measurements?

Technical comments

- Page 3, line 77, Introduction : "... Section 2 introduces the reference wind farms, the wind farm and the atmospheric data...".
   The sentence should be read again: what are the reference wind farms? What is the wind farm?
- Page 14, line 261 : "... in both figures are ..." Write explicitly in Figs 8 (a) and 8 (b).

---

## Author Comment (AC1)

**Author's response - The impact of far-reaching offshore cluster wakes on wind turbine fatigue loads**

Arjun Anantharaman, Jörge Schneemann, Frauke Theuer, Laurent Beaudet,
Valentin Bernard, Paul Deglaire, and Martin Kühn

We would like to thank the two reviewers for taking the time and effort to give detailed and constructive feedback. We believe the implemented changes have improved the quality of the manuscript. Please find below our responses to all of the comments along with the changes in the manuscript, where applicable.

- Red - Reviewer comment

- Black - Response to the reviewer from the authors

- Blue - Changed text in the manuscript

**Reviewer 1**

**Major comments**

1. The reliance on nacelle fore-aft acceleration as a fatigue load proxy, while validated against limited measurements and simulations, raises concerns. The correlation coefficients indicate moderate predictive power, which may not fully capture complex load dynamics. Please include additional load proxies (e.g., tower base bending moments or blade root strains) or validate fore-aft acceleration against direct strain gauge measurements from the studied turbines. Discuss the limitations of using a single proxy for fatigue assessment, particularly in waked conditions.

We thank the reviewer for bringing up this point. The turbines in the A/HS wind farm for which the load effects were analysed were not fitted with load sensors, and this is the reason we looked for a suitable SCADA variable that could act as a proxy for loads. The standard deviation of the fore-aft nacelle accelerations have been shown to be a good proxy for the relative change of fatigue loads, not only in the current manuscript, but also in several studies when comparing different SCADA signals [Cosack, 2010, Mittelmeier et al., 2017]. The correlation coefficients indicate moderate predictive power only between the load proxy $a'_{\mathrm{fa}}$ and the other SCADA variables, but show high correlations when compared to Damage Equivalent Loads (DEL) or actual load measurements (Please refer to Table 3 in the manuscript).

In Section 3.1 of the manuscript, we delve into the details on why we chose the $a'_{\mathrm{fa}}$ as the load proxy and compare it to turbines with strain gauges. In the Table 3 of the manuscript, we have correlations between the load proxy and both blade and tower strain gauges from the Borssele wind farm. This wind farm has the same turbine type as the one in the current study, with the exception of a higher rated power but with the same rotor diameter and hub height. All the correlations show good relationships and justify the choice of $a'_{\mathrm{fa}}$ as the load proxy in our current study. We also compared the load proxy for the exact same turbine in numerical simulations with the aero-elastic solver BhawC and found mostly good correlations. The simulations considered turbulent wind fields in accordance with typical design calculations rather than emulated site specific wake situations. Nevertheless, we see a high correlation between $a'_{\mathrm{fa}}$ and the blade root flapwise moments and the tower top yaw moments.

We agree with the reviewer that the load proxy cannot fully capture the complex dynamics within the cluster wake, and we have discussed this in Section 5, where we highlight the drawbacks of our analysis. In section 5.2, we specifically address the point that the SL tables, which are used to quantify loads, do not perform well in waked situations. This can be seen in the relatively poor performance of

the $\zeta_{\mathrm{inner}}$ parameter, which has higher uncertainties. We have added some text to the main body and discussion in conjunction with the comments from both the reviewers.

Section 3.2:
"We also created the SL-inner table (not shown) for inner farm wake conditions and found that the SEM was significantly higher than SL-free, due to lower data and more complex flow dynamics due to inner farm wakes in each $u$ and $u'$ bin."
Section 4.1:
"For this case study, the cluster wake does not have any adverse load effects and even has lower $a'_{\mathrm{fa}}$ than the free-wind turbines (it is not constant along the row like the SL tables), pointing to an over-estimation bias of the SL tables. This will be explained further in Section 4.3."
Section 5.1:
The load proxy also does not perform well when the wind speed fluctuations are higher, which occurs when there is increased atmospheric turbulence due to inner farm wakes. This is apparent from the higher biases in the $\zeta_{\mathrm{inner}}$ parameter and also in the SL-inner tables (not shown), which has higher SEM values as compared to SL-free.

2. The use of 10-min SCADA averages may obscure high-frequency load fluctuations critical for fatigue analysis. Additionally, turbulence intensity (TI) derived from nacelle anemometers is inherently biased due to rotor interference, as acknowledged but not sufficiently corrected. Please incorporate high-resolution (e.g., 1 Hz) SCADA data for all analyses, not just spectral studies. Apply rotor-induced turbulence correction models to improve TI estimates.

We agree with the reviewer that 10-min SCADA data may obscure high-frequency load fluctuations that could affect fatigue analysis. However, we only have high-frequency (10 Hz) SCADA data for 56 out of the analysed 96 cluster wake scenarios, and so we made the choice to focus only on the larger time scale impact of the cluster wake. Our aim with this manuscript is more to use commonly available 10-min SCADA data to assess the persistent effects of the cluster wake on offshore wind turbines, which we believe was achieved with the data utilised. This study also only serves as a proof to the existence of a load effect due to cluster wakes. We have recommended that the analysis of transient effects (with lots of high-frequency SCADA) of the cluster wake's impact should be the focus of future research.

We have added the following paragraph to Section 5.2:
"Additionally, high-frequency SCADA data could be used for analysing the wind turbine response to the transients at the moment the cluster wake impacts the downstream wind farms. Cluster wakes are inherently complex flow structures, and using only 10 min SCADA could leave out crucial information on the flow dynamics that could adversely impact fatigue life of offshore wind turbines."

We agree with the reviewer that the TI calculated from the nacelle anemometers is inherently biased. This is also why we do not use the nacelle TI to draw any conclusions, but rather only use the standard deviation of the wind speed to create the SL lookup tables. While the wind speed measurements (also its standard deviation) from the nacelle anemometers could be improved with better rotor correction models, all our analyses rely on the load proxy to quantify the relative change of fatigue effects. We acknowledge that the correction models are also more complex and are atmospheric stratification-dependent [St Martin et al., 2017], which we mentioned in our discussion (Section 5.2). We believe that further analysis in TI correction models would be beneficial in future research focusing on load analysis from cluster wakes as a function of the ambient TI .

3. The reported 2.5% load increase in cluster-waked turbines lacks statistical robustness. With only 96 cases (56 for spectral analysis), the sample size may be insufficient to generalize findings, especially given the high variability in offshore conditions. Please perform hypothesis testing (e.g., t-tests or ANOVA) to confirm the significance of differences between cluster-waked and free-wind turbines. Expand the dataset to include more cases across seasons and stability regimes.

We thank the reviewer for bringing to our attention the lack of proof of the statistical significance results in our manuscript. We have now conducted both the two-sample t-tests and unequal ANOVA tests, both showing that the current conclusions are statistically significant. Before we present the results, we would like to clarify the size of the dataset and why the results show statistical significance. While there are only 96 cluster wake cases, within each case, there are several turbines which are

considered in the calculation of the load effect using the $\zeta_{\text{free}}$ parameter. The turbines selected in each scenario are detailed in Section 4.2 of the manuscript. This means, that for the 96 cluster wake cases, we are actually computing the load proxy (using reference conditions) and $\zeta_{\text{free}}$ for 1180 turbines, each with a unique wind speed $u$ and fluctuation of wind speed $u'$. For these turbines within the cluster wake, we compare the distribution of values to the turbines in free wind, and perform the two sample t-test and unequal ANOVA test (since two variables are of different sizes) and obtain the following results.

Two-sampled T-test:
P = 0.012 which is less than 0.05, for the 5% significance level
H=1, implying a rejection of the null hypothesis
The above is assuming that both variables have equal variances, but we account for that and redo the t-test (Welch with unequal variances).
Welch's t-test, same as Anova for two groups:
P =0.004 which is much less than 0.05, for the 5% significance level
H=1, implying a rejection of the null hypothesis

We also performed the Anova test (using the f-test) and obtained the same results, showing that the presented results for the increase in loads for cluster wake-affected turbines is statistically significant. Additionally, we also found that the parameter $\zeta_{\text{inner}}$ does not show statistically significant results, and this was also expected since the uncertainties were much higher in waked situations, pointing to more data needed in SL-inner reference situations. We have added this to the discussion section of the manuscript.

"We also performed statistical tests on the distributions of $\zeta_{\text{free}}$ when comparing free wind turbines and cluster wake-affected turbines to determine if the presented results showed statistical significance. We used both the two-sided t test and the Welch's Anova test [Wackerly, 2008], while the latter does not assume the two variables having equal variances. It is to be noted that while there are only 96 cluster wake situations, the $\zeta_{\text{free}}$ parameter is calculated on a turbine basis (see Sec. 4.2). This means, for example that $\zeta_{\text{free}}$ for cluster wake-affected turbines actually contains 1180 values. The results from both tests show $p$ values less than 0.05 for the 5% significance level and a conclusive rejection of the null hypothesis. This means that the presented methodology to quantify load effects shows statistically significant results."

Unfortunately, we do not have the possibility to expand the dataset across seasons and stability regimes. However, with our results showing statistical significance, we strongly believe that it can be used as a motivation for further research across wind farm cluster and over several years and that the presented results support our conclusions.

4. The conclusion that atmospheric stratification has "no impact" on loads contradicts prior studies showing stability-dependent fatigue effects. The simplified stability classification (3 regimes) and reliance on WRF-derived Monin-Obukhov lengths may oversimplify boundary layer dynamics. If possible, please re-examine stability effects using direct measurements (e.g., lidar-derived TI or temperature gradients). Consider finer stability classifications (e.g., 5–6 regimes) to capture subtle interactions between wakes and stratification.

The effect that atmospheric stratification has no impact on loads is a broad statement, and we would like to clarify as to when this is applicable. From Figure 13 (in the manuscript), across all stability classes, the cluster wake-affected turbines show higher loads than turbines in free wind. The magnitude of this increase is highest in neutral stratification (5%), but this stability class also has the lowest number of cases. We agree with the reviewer that the simplified stability regimes may simplify boundary layer dynamics. We also used finer stability regime classifications, but this significantly reduced the number of cluster wake cases in each stability class. Figure 1 shows the results obtained using finer stability classifications and Table 1 shows the calculated parameter values.

When we divided the stability classes into finer regimes as shown below, there was no trend for the increase in loads vs atmospheric stability. We agree that this results in non-conclusive stability effects of cluster wakes on loads, and so we have reformulated the conclusions to reflect the same. Despite

[Figure]

Figure 1: Distribution of the $\zeta_{\text{free}}$ parameter shown as histograms for five stability classes. The number of cluster wake cases for each stability class is indicated in brackets in the axis label.

this, the consistent increase in the load indicator $\zeta_{\text{free}}$ for cluster wake turbines supports our overall conclusions. We took the decision to keep the simplified three stability classification to ensure a higher number of cases within each stability regime.

Table 1: Stability classification in finer divisions, adapted from Araújo da Silva et al. [2022]

| Stability | Monin Obukhov length $L$ [m] | Number of cluster wake cases |
|---|---|---|
| Unstable | $-200 < L < 0$ | 30 |
| Weakly unstable | $-1000 < L < -200$ | 30 |
| Near neutral | $|L| > 1000$ | 4 |
| Weakly stable | $200 < L < 1000$ | 9 |
| Stable | $0 < L < 200$ | 23 |

We added the following statement to section 4.3:
"We also tried finer classifications of atmospheric stability, and did not find any noticeable differences compared to the more coarse classification presented here."

Previous studies have indicated that stability effects influence fatigue loads, and also the cluster wake development itself, as we mention in the introduction. Based on our dataset, we cannot conclude any significant effect of the stability on fatigue loads. This could be attributed to the magnitude of the load increase due to cluster wakes, which is very small (2.5%) as compared to the wake effects of adjacent turbines within the wind farm ($\approx$20%). Additionally, more data could definitely help in drawing stronger conclusions. Unfortunately, we do not have access to lidar or met-mast based TI/temperature measurements for the calculation of stability or data across more seasons. However, we did a validation campaign from lidar measurements and temperature sensors against WRF measurements, and found a very good match, so we are confident that the utilized stability values are meaningful for the current analysis. We also modified the objective statement to not include atmospheric stratification as it is only a small part of our overall analysis. (see specific comment#1 from Reviewer#2).

5. The study isolates cluster wakes but does not address potential superposition with inner-farm wakes, which could amplify load effects. The comparison between "last-row" and inner-farm turbines is superficial. Please analyze combined wake scenarios (cluster + inner-farm, e.g., Journal of Cleaner Production 2023, 396: 136529) to assess cumulative load impacts.

We agree with the reviewer's comment that we did not address in detail the potential super-positional effects of cluster wakes with inner farm wakes. The study [Wang et al., 2023] uses WRF to determine combined power losses on new wind farms and the outcome focuses on validation of the advanced WRF model with ERA5 reanalysis data. Their analysis points to wakes extending as far as 100 km, dependent on the normalised power of the upstream wind farms. Our current analysis is for moderate wakes distances, and for combined super-positional effects, the absolute value of the load effects should be considered.

Table 4 from the manuscript already gives an indication that the cluster wake causes on average 15% reduction in the wind speed (which is further reduced inside the wind farm, last row included) and a corresponding 7% reduction in the load proxy. The front-row turbines within the cluster wake experience much lower loads than if the cluster wake was not present. This is due to the strong dependence of the load proxy on the inflow wind speed. The reason why we considered the last row specifically, is because it contains the superposition of all inner farm wakes, and for turbines within the wind farm, we already take the superposition of inner farm wakes into account. This means, that despite the cluster wake impacting the wind farm, the load effects are in fact lower for the front row, and this does not carry over inside the wind farm, at least based on the current data set. We also analysed the last row separately, which we mention in manuscript but do not elaborate on, since the outcome was that there were no additional effects of the cluster wake within the wind farm. It is for this reason that we do not analyse the last row with reference loads, as it would not shed light on super-positional effects. In the manuscript we wrote in Section 4.2:

*"We also show the effects on the last row of turbines, which are impacted by the aggregation of all upstream turbine wakes, and the qualitative results match the inner-farm turbines, which would experience only the wakes of a few turbines. We also made a comparison of the turbines in the last row for SSW wind directions and found no increase in $a'_{\mathrm{fa}}$ when splitting the last row into halves: one containing only inner-farm wakes and one containing the additional cluster wake caused velocity deficits. Though the turbine layout affects the propagation of inner-farm wakes, the similar effects on the inner-farm and last-row turbines in Table 4 indicate no combined effects of the cluster wake on inner-farm turbines."*

We have already demonstrated that capturing wake effects (inside the farm) is extremely challenging and creating a lookup table introduces high errors due to the complex flow dynamics (cf. comment#17 from Reviewer 2 for the SL-inner values). This is why we did not consider the last row turbines for calculating reference loads, and also why the $\zeta_{\mathrm{inner}}$ parameter does not show statistically significant results. The three-dimensional flow within a wind farm cannot be captured by one load proxy alone, notwithstanding additional effects from cluster wakes. We also conferred as to the current standard practice in the industry on what loads are considered crucial for fatigue assessment within this context. The stated 2.5% would be only concerning if it occurred in absolute conditions, and since the lower wind speeds within the cluster wake have a load-reducing effect, the current study does not necessitate a rethink of design standards, at least not without more analysis in more wind farm layouts and operational conditions.

**Minor Concerns**

1. The discussion omits recent advances in cluster wake modelling and fatigue load prediction using machine learning. Update references to reflect state-of-the-art methodologies.

We thank the reviewer for bringing this to our attention. We have added more studies focussing on machine learning-based fatigue load analysis in our introduction section. Additionally, based on comments from reviewer#2, we also added some references on numerical modelling of cluster wakes.

"There are also several numerical studies on modelling cluster wakes with the focus on accurate estimation of production losses and wake deficit estimation. A state-of-the art review of numerical studies on offshore cluster wake modelling can be found in Ouro et al. [2025]."

"SCADA data has also been utilised in data-driven machine learning models to predict fatigue behaviour of turbines within a wind farm. These studies show promising results that SCADA and acceleration data can be used to estimate long-term fatigue damage [de N Santos et al., 2023, 2024] with potential for lifetime-based decision making for future wind farms."

2. The manuscript understates the operational relevance of findings. Elaborate on how the 2.5% load increase translates to lifetime extension or maintenance strategies.

The standard practice in industries is to consider an omni-directional TI: this means the design for wake-induced load effects is based on a worst case scenario for a turbine that would face the most load effects from inner farm wakes. This also relates to comment#5 , wherein we show that with the current analysis, the increase in loads is only relevant when the wind speed reductions are considered. The absolute effects of the cluster wake are not load-increasing, rather load-alleviating due to the lower wind speeds within the wake. For our analysed cluster wake, the current strategies are well equipped, but this could be different for a different wind farm layout and atmospheric conditions, which we address in the recommendations in Section 5.2. We added some further information regarding future studies and applicability to other wind farm clusters.

"The calculated load increase in reference conditions is also limited to the analysed wind farm cluster, and the effects on loads will depend on not only the cluster layout, but also the operating conditions of turbines in both clusters."

3. This work provides a foundational exploration of cluster wake impacts on fatigue loads but requires methodological refinements and expanded datasets to strengthen its conclusions. Addressing the above concerns will elevate the study's scientific rigor and applicability to wind farm design standards.

We would like to once again thank the reviewer for the detailed critique, based on which we refined the conclusions to better suit the utlised dataset. We added statistical tests to strengthen the conclusions and prove the significance of the load increase due to cluster wakes. We also would like to re-iterate that from the wind farm operator's perspective, the loads due to cluster wakes are lower on absolute terms due to the reduction in wind speed and so the current design standards are sufficient to account for cluster wakes. While expanding the dataset is not feasible within the current study, we are confident that future studies can use the presented methodology to further analyse cluster wakes and potentially lead to re-evaluation of design standards in specific scenarios where there are higher loads due to persistent cluster wakes at different inter-farm distances and layouts.

**Reviewer 2**

**Specific comments**

1. Page 3, lines 65-66, Introduction : "Our objective is to determine if far-reaching cluster wakes impact individual downstream wind turbine short -term fatigue loading dependent on the atmospheric stratification.". I would adapt/modify this sentence as the study of dependence on atmospheric stratification is only a small part of the analysis.

We agree with the reviewer that while we do assess the effect of atmospheric stratification on cluster wake-caused loads, it is not a major aspect of the paper and so we have modified the objective statement to reflect the main aim of the paper, which is not only limited to the stratification, but in larger part the wind speed and its standard deviation.

"Our objective is to determine if far-reaching cluster wakes impact individual downstream wind turbine short-term fatigue loading dependent on the atmospheric conditions."

2. Page 4, line 94, Section 2.1 : "... data from the sister wind farms Albatros and Hohe See...". The authors should also refer to Fig 2 to help to the reader locate wind farms of interest in the N8 cluster.

We agree with the reviewer that there has been no clarification on the differences in the Albatros and Hohe See wind farms in any of the presented layouts. We have now modified Figure 2(a) and differentiated between the sister wind farms and also added a clarifying note in the caption, since this demarcation is not made in the rest of the manuscript as we only refer to the combined wind farms as A/HS .

"We use operational SCADA data from the sister wind farms Albatros and Hohe See in the N-8 cluster (see Fig. 2a) in the period from..."

3. Page 4, lines 103-104, Section2.1 : "we chose turbines in free-wind (green) and inner-farm wake (red) as reference conditions...". At this stage of the reading, the meaning "reference conditions" is not clear. This will become clearer later, but for the sake of readability, this should be explained a little more here.

The sentence was re-written with better clarifications as to why these turbines were chosen for the same wind direction sector and removed the wording of reference conditions.

"The front row of A/HS is defined for this work as the 22 turbines which are directly impacted by the N-6 cluster wake, highlighted blue in Fig. 2b. To simultaneously compare the cluster wake-affected turbines to both turbines in free-wind and inside the wind farm, we fixed these turbines for the cluster wake wind direction sector. These are referred to as free-wind and inner-farm wake turbines (green and red markers respectively, in Fig. 2b) for the same 230° to 270° wind direction sector."

4. Page 4, line 105, Section2.1 : "The wind direction was derived from the 10-min mean nacelle positions". Which turbines were used to calculate the wind direction? All turbines?

We calculated the wind direction from the mean nacelle position of the front row turbines and not all turbines within the wind farm, since these front row turbines were mainly used in analysing cluster wake effects. We also separately computed a wind rose from all the turbines and found it to be similar. We chose to keep only the front row turbine based wind direction for our analysis. The caption of Figure 3 was also adjusted to reflect the same.

"The wind direction was derived from the 10-min mean nacelle positions of the front row turbines, and the resulting distributions for the nacelle positions for normally operating turbines in A/HS (front row)..."

5. Page 5, lines 114-115, Section2.2 : the authors wrote that they filtered out situations where wind speed was higher that the rated speed. What is a situation with above rated wind speeds: is it when all upstream turbines are impacted by an above-rated wind speed? The authors should clarify this, especially as later in the article, they write that if turbines in free wind speed are in above rated conditions, that is not a problem (see comment 8) And why do the authors discard situations where the wind speed is higher than the rated speed?

We thank the reviewer for pointing out this specifically, since we did not clarify why we allowed cases

where the wind speed was higher than the rated wind speed for turbines in free wind. The SL tables were created to tabulate loads in reference conditions for below rated wind speeds, where there are no load mitigation measures such as pitching in effect. While this might also be an interesting wind speed regime to look at, this would require a much deeper look into the behaviour of the turbine for different pitch angles, which we believe is out of scope for this paper.

For the comparison of the cluster wake-caused loads, a turbine inside the cluster wake is considered, and the performance indicator $\zeta$ is calculated only for this below rated condition of the turbine. It is for this reason we consider cases where the free wind turbines may be above rated wind speeds and pitching, but the turbines of interest (inside the cluster wake) are still operating in below rated wind speeds due to the significant reduction in wind speed caused by the cluster wake. We ensure that there were no pitching effects in any of the cluster wake-affected turbines by manually examining these cases. We added a clarifying sentence in this part to highlight where an exception occurs.

"To limit our analysis to cases with turbines in normal operation we filtered out the data in situations of curtailment, turbine maintenance, shut down, and wind speeds below cut-in and above-rated. We also consider a cluster wake case as valid only when the turbines within the cluster wake are operating in below-rated wind speeds, although a few turbines in free-wind might be above-rated. This is because we require all the analysed cluster wake-affected turbines to be in normal operation below rated wind speeds."

6. Page 7, Figure 4 (a): I guess the authors do not have a snapshot a little later, to really see the wake of the cluster that will impact the downstream wind farms. The snapshot should be explained a bit more, to be useful in the analysis/paper. Do we see in figure 4 (a) a part of the cluster wake that will have an impact on cluster N8?
The satellite SAR snapshot is only once per day over that region, and our aim was to only show proof of the cluster wake's existence. We mention this once more when describing the exemplary wake case in Section 4.1. While the case study is selected from 06:40 onwards, the wind direction indicates that even at 05:57 (time of SAR snapshot), for 270°, the cluster wake would still hit the N-8 cluster. Unfortunately, we do not have images of the wake during the exact time period of the selected wake case or after it. In Figure 4a, we already see the outline of the cluster wake, which is shown along the wind direction arrow. The arrow has been updated to reflect the wind direction at the SAR timestamp, rather than the time of the selected cluster wake case (270° as opposed to 261°).

Figure 4a displays a wind field derived from a satellite SAR snapshot (ESA, 2021) shortly before the cluster wake situation. Fig.4b shows the time series of the nacelle position and wind speed for the exact exemplary wake situation. The wind direction in the SAR snapshot is 270°, and even though this exact time is not considered for the selection of a case (since the wind direction has not yet stabilised), the cluster wake is visibly impacting the A/HS turbines.

7. Page 7, Figure 4 (b): Are these wind speed and direction for each wind turbine?
The wind speed and direction and the mean of the front row turbines. We have mentioned this in the figure caption and also added clarification in the text, see comment #4.

8. Page 7, lines 124 to 127 : "The wind speed in the front row of turbines which are not affected by the cluster wake are also within 0.5 m/s of each other". Is it a condition for the selection of a cluster wake case?
This is not an explicit condition to be considered a cluster wake case, and we mentioned this only to emphasize that the mean wind speed is nearly constant across the front row, serving further to have a nearly constant wind speed and direction for the duration of the cluster wake, specifically for the turbines in free wind. Upon reading this sentence again, we found that this only makes it more difficult for the reader to interpret, so we removed this sentence.
And I do not understand the sentence that follows (same paragraph): "There are four cluster wake cases (out of 96) where this does not hold true since the free-wind turbine are operating above the rated wind speed." What does the term "this" refer to? Why does operating at a wind speed higher than the rated speed necessarily mean that the wind speeds of the free-wind turbines are not within 0.5 m/s of each other?
We agree with the reviewer that the wind speeds being 0.5 m/s of each other is not affected by free

wind turbines being above rated, so we removed this sentence. "this" referred earlier to the 0.5 m/s mean wind speed for the cluster wake case, and we found that this added no value to the analysis.

Answered in Comment #5. Additionally, the main findings of our paper relate to increased loads using the SL tables, and in all the calculated $\zeta$ values, the wind speed is below rated (since these are calculated for cluster wake affected turbines). When we used the $\zeta$ parameter for turbines in free wind, the calculated bias was less than 0.8%, hence these turbines were not discarded in the final results.

9. Page 7, lines 131-132 : the authors mention the power deficit caused by the cluster wake in the first line, but they do not detail how they calculate it. Is it calculated in the same way as the power deficit discussed in Section4.2? The definition of power deficit should be detailed here, and perhaps recalled later in the results.
We defined the power deficit in the sentence preceding the comment.

"Further, we checked the difference in power of the front row of the N-8 cluster (A/HS) by computing the power deficit $\Delta P$ as the difference in power between the maximum and minimum power producing turbines. Only cases where the magnitude of the power deficit caused by the cluster wake in the front row, $\Delta P$ , was higher than ...".

10. Page 8, Section2.3, second paragraph: If I understood correctly, the authors validated WRF simulations based on atmospheric measurements for another period, and they found a good match. So they use another WRF simulations for the considered period. Are there the results of Canadillas et al. ? If yes, the authors should write that it comes from this reference explicitly and maybe detail the reference in one sentence.
The WRF simulations carried out for validation with meteorological measurements were indeed from a different set of simulations [Dörenkämper et al., 2020]. We thank the reviewer for bringing this to our attention. We have added the reference accordingly.

"We compared WRF stability values obtained from simulations carried out using the setup of the New European Wind Atlas (NEWA) [Dörenkämper et al., 2020] against atmospheric measurements, ..."

11. Page 9, lines 157-160, Section3: This sentence is difficult to understand. Is this what you need to do for a complete fatigue analysis?
We describe how the usual fatigue load calculations are done using 10-min data or from spectral information. We mention this specifically since we are commenting on the short-term fatigue response of the turbines to the presence of the cluster wake instead of the usual fatigue calculations for standard load cases under IEC norms and rain flow counting of time series. We also show the spectra from a load proxy, not a strain gauge. The sentence that follows "We limit the comparison.." also serves to reinforce the aim and scope of the fatigue load analysis of the current manuscript. We have reformulated the sentences to improve clarity.

"Wind turbine fatigue loads are typically compared by load spectra or Damage Equivalent Loads (DEL) derived from rain-flow counting of load time series. These load calculations are performed according to IEC 61400-1 [2019] for different site and operational conditions. This comparison is done for either short-term fatigue loads (10 min) or for long-term fatigue loads, which are extrapolated to design lifetime based on the frequency of occurrence of certain load situations. We limit the comparison to the analysis of short-term load fatigue loads in different atmospheric conditions."

12. Page 9, lines 171-173, Section3.1 : it is written that the nacelle accelerations have an approximately linear relationship with wind speed below the nominal value. What happens for wind speeds higher than the rated speed? (Because the authors wrote that they kept situations where the free wind turbines are higher than the rated speed).
Free wind turbines are not considered at all for the analysis of effects in cluster wake situations, as we use the SL tables at normalised wind speeds, which occurs below the rated wind speed of the turbine. The load proxy $(a'_{\mathrm{fa}})$ does continue to increase in a similar manner as below rated, but we chose only below-rated as mentioned previously to avoid load reduction due to pitching of the turbine blades.

We mention an approximately linear relationship since the correlation between the $a'_{\text{fa}}$ and both the wind speed and its fluctuation ($u$ and $u'$) is around 0.5. This low correlation is also affected by the position of the anemometer, as the rotor disturbs the flow. Additionally, there is also signal noise, and in keeping the analysis to below rated wind speeds, our aim is to reduce as many uncertainties that could affect the conclusions.

13. Page 10, line 181 : for the Borssele wind farm, what period is used to compute the DELs? Do the authors compute correlations over a wide range of atmospheric conditions and stratifications? I assume that correlations are better for certain wind conditions... It would be useful to have more details about these calculations. Same comment for the numerical simulation. Further in the paper (Section5.1, discussion), it is written that this is a numerical simulation for onshore turbine. It should be mentioned here.

The strain gauge measurements from the Borssele wind farm were utilised for a period of seven months from 01-Jan 2021 till 05-Aug-2021. We have added this information to the manuscript. Over this limited time period, no specific stratification information was available. For the aeroelastic simulations, we used data from the same turbine type as in the A/HS, but from the design stage for an onshore prototype. We have added some information here regarding both data sets.

"Around seven months of measurement data (01-01-2021 to 05-08-2021) were used to derive these correlations."

"It is to be noted that the simulations were taken from an onshore setup and exclude any potential hydrodynamic effects on the loads. There is also no stratification input in the simulations".

14. For the DEL, do the authors correct the number of cycles based on the mean loads?

The strain gauge measurements from the Borssele wind farm were processed from the time series in 10-min DEL values with a suitable slope of the S-N curve and 600 reference cycles. The number of cycles were not corrected here based on mean loads, but we only use the correlation between the time series of the load proxy and the DEL variables. Even if there was a bias, the correlations would remain unaffected.

15. Page 10, Table 3 : How do the authors explain the difference in correlations between measurements and numerical simulations (for the tower top tilt moment)?

The correlations between the load proxy and the DEL variables are lower in general for the simulations, this is especially true for the tower top tilt moment. One potential reason for this could be due to the nature of the simulations, which do not consider hydrodynamic effects such as waves and currents on the turbine structure. These effects cause markedly more loading effects and different behaviour as compared to onshore turbines [Damiani, 2016]. The simulations were run for the same turbine type but neglecting these effects could cause much lower response in some of these variables. Additionally, it could be that the motions are driven in reality by the turbulent structures in the wind field, which is not captured in ideal simulation scenarios. Our aim in showing this variable in particular is to highlight the differences between simulations and measurements, and to demonstrate that despite differences, our choice of load proxy holds good in both situations.

16. Page 11 : I do not understand the difference between u' and the anemometer-based TI. How are they computed respectively? The anemometer-based TI is computed based on u' no? Why is the correlation between a'fa and u' good, but not between a'fa and TI?

Nacelle anemometer TI is $u'$ divided by $u$ for the same 10-min period from SCADA. The standard deviation of the nacelle wind speed $u'$ corresponds to the disturbed flow behind the rotor, and when divided by the mean wind speed the physical relationship is lost to the calculated anemometer TI. This is because the correlation between $u$ and $u'$ is low to moderate (R=0.37 for free wind and 0.59 for inner farm-wake) depending on the inflow conditions. The disturbance of the flow due to the rotor also affects the fluctuations in wind speed across each 10-min interval. This already gives a first indication that the $u'$ is not a good representation of the nacelle-based TI, let alone the ambient atmospheric TI. All of these factors are reflected in the correlation between $a'_{\text{fa}}$ and TI, which is poor to low (R=0.09 to 0.29). This is also why we built the SL tables as a function of $u$ and $u'$, not $u$ and TI.

[Figure]

[Figure]

Figure 2: (a) Standard Loads (SL) matrix for $a'_{\mathrm{fa}}$ in inner farm wake situations and the corresponding SEM (b), both dependent on $u$ and $u'$.

17. Page 12, Figure 7 : It would be interesting to add SL for inner farm effects even if it is not explained in detail. (but this would increase the size of the paper)

We thank the review for noticing that the SL for inner farm effects was left out. We initially did have the SL inner matrix image in the manuscript (cf. Figure 2), but we left it out as it increased the size of the paper and also did not add to the conclusions. The main performance indicator with low biases was $\zeta_{\mathrm{free}}$ and so we decided to leave the image of the SL for inner farm wake conditions out of the manuscript. In Fig. 2 the SEM is noticeably higher in many more of the bins as compared to SL free, which is also the reason why we stated that for inner farm wake situations, much more data is needed since the flow is very complex and directionally dependent. We added some text to address this in the methods section, also in conjunction with comments from reviewer#1.

"We also created the SL-inner table (not shown) for inner farm wake conditions and found that the SEM was much higher than SL-free, due to lower data and more complex flow dynamics due to inner farm wakes in each $u$ and $u'$ bin."

18. Page 14, lines 270 : "..., possibly due to the combined effects of increased wake recovery in very unstable stratification and lower wind speeds due to the cluster wake". I do not understand this sentence: the effects are opposite. Why would a combination of these effects lead to a decrease in loads?

We agree with the reviewer that a combination of these effects would not necessarily result in a load decrease. We have removed this sentence and added more clarification, both in the case study section and also later where the stability comparison is made for the performance indicators. Both changes are shown below: (same text change as comment#1 from reviewer#1). The SL tables show biases across all stratifications, though it is marginally higher in stable situations potentially due to the lower number of cases. This means that for more accurate stability-based case study analysis, the SL tables have to also be binned according to stability, which would be a very interesting topic for further research.

"For this case study, the cluster wake does not have any adverse load effects and even has lower $a'_{\mathrm{fa}}$ than the free-wind turbines (it is not constant along the row like the SL tables), pointing to an over-estimation bias of the SL tables. This will be explained further in Section 4.3."

Section 4.3:

"The uncertainties are slightly higher for neutral and stable stratification than unstable stratification. This is directly related to the number of cluster wake cases in each stability class, as seen in Table 2."

[Figure]

Figure 3: Layout of alpha ventus, FINO1, and the surrounding wind farms including relative directions and distances. The origin is the FINO1 platform. Background layer used with permission from 4C Offshore (last access: 1 December 2020). Image taken directly from Pettas et al. [2021].

19. Page 21, line 373 : the authors write that, for 56 out of 96 cases, spectra were available. What does this mean? Because the authors wrote on page 20 that they averaged the turbine spectra for all cases: so, "all cases" means 56 cases or 96?

We only had high-frequency SCADA data for 56 out of the 96 cases, and in the context of the sentence, we meant the averaged spectra for all 56 cases, which we have now added for clarity.

"We then averaged the spectra for these turbines across all the 56 cases with high-frequency data, and the result .."

20. Page 22, line 386 : "Pettas et al. (2021) found the maximum distance that the wakes from neighbouring wind farms impacted SCADA signals was 6.5 km". Whatever the size of the upstream wind farm? Or for one layout/situation?

In the work of Pettas (2021), they studied the effect of wind farm wakes by using a met mast and one turbine of the alpha ventus wind farm and analysing the effect of all the wind farms that were eventually built around it, at varying distances. In this sentence, the maximum effect that they could detect an effect of the neighbouring farm was 6.5 km. The wind farms are highlighted in Figure 3 taken from their publication. For the specific wind farms in question, it would be the effect of the upstream wind farm Trianel Borkum 1 (2015) on the alpha ventus wind farm, which is at a minimum distance of 6.5 km away. We added some text in the manuscript that describes the size of this wind farm and compares the scale of their study to ours. We have partially rewritten this paragraph also taking into consideration comment#21.

"The upstream wind farm in their study (Trianel Borkum 1) has 40 turbines, and a total wind farm rated power of 200 MW. We only considered this situation as a point of comparison as further cluster wake scenarios in their study had low inter-farm distances (< 3 km)."

21. Page 22, lines 388-389 : "... that showed increased values due to the cluster wake were the pitch and the generator speed. Firstly, we found a small increase in the nacelle TI..." I do not understand the link, and I think that "Firstly" is not the right term to begin this sentence. Do the authors mean

that Pettas et al. found nothing at 6.5 km although they found impacts at 15 km ? It should be clearer. And next lines (lines 389-390) : the nacelle anemometer is located at the same position, whatever the turbine (in free- wind or in cluster wake). So why might the increase of TI for cluster wake turbines be attributed to the location of the nacelle anemometer?
We rephrased the sentence. We meant to state that Pettas et al. [2021] found no DEL variable response at 6.5 km, but only weak pitch and generator rpm responses. It is to be noted that in their study they did not account for low wind speeds in the cluster wake and showed absolute values for their results. For the nacelle anemometers, we meant to state that the increase in nacelle TI does not mean an increase in atmospheric TI, since this increase is much lower than the corresponding increase for inner farm wakes affected turbines. (16% as compared to 90%, in Table 4 of the manuscript). This is why we attribute it to signal uncertainties and are cautious in drawing any conclusions with respect to the atmospheric TI change caused by cluster wakes at these downstream distances. Rewritten paragraph:

"They found that the DEL did not show any significant changes at 6.5 km downstream distances, and the only parameters that showed increased values due to the cluster wake were the pitch and the generator speed.

Comparatively, the upstream wind farm cluster in our study has a total rated power of 1062 MW. We found a small increase in the nacelle TI for the turbines in the cluster wake more than 15 km downstream, shown in Table 4. This increase in the absolute value of nacelle TI is far lower than the corresponding increase due to inner farm wakes. Additionally, we also found that the fluctuations of the generator speed and turbine power increased in cluster wake-affected turbines, but the magnitude of the increase was much lower as for the turbines inside the wind farm. We consider the smaller increases in nacelle TI, generator speed fluctuations and turbine power fluctuations to be partially affected by signal uncertainties. We recommend more data over longer periods be analysed to definitely conclude on a potential effect in these variables for cluster wake-affected turbines."

22. Page 24, discussion: are there any numerical studies of cluster wakes, which might help the authors to explain some of their results/measurements?
To the best of our knowledge, the numerical studies on cluster wakes are focussed primarily on the improvement of the models (WRF, fast engineering models, etc.) to match wind speed/turbine power measurements [Anantharaman et al., 2022] or on predicting power losses in conjunction with other large scale effects such as global blockage [Nygaard et al., 2020]. A comprehensive review paper published recently summarises all the work on numerical modelling of cluster wakes, and all their conclusions are focussed on power losses and wind speed deficits [Ouro et al., 2025]. It is our hope that our manuscript will motivate researchers to also look at the loading effects of cluster wakes on existing and upcoming wind farm clusters. We have added some references regarding numerical modelling and machine learning-based load predictions to the manuscript, please refer the answer to minor comment #2 from Reviewer#1.

**Technical comments**

1. Page 3, line 77, Introduction : "... Section 2 introduces the reference wind farms, the wind farm and the atmospheric data...". The sentence should be read again: what are the reference wind farms? What is the wind farm?
Sentence changed to improve clarity.
Section 2 introduces the reference wind farms (affected by the cluster wake), the wind farm and atmospheric data used in the analysis and highlights the steps involved in choosing a cluster wake case.

2. Page 14, line 261 : "... in both figures are ..." Write explicitly in Figs 8 (a) and 8 (b).
Changed the sentence as recommended.

**Changed figures in the manuscript**

[Figure]

**Figure 2.** (a) Region of the N-6 cluster (dots) and the N-8 cluster (filled circles, filled triangles and empty circles). The distinction between the Albatros and Hohe See wind farms in the N-8 cluster is only shown in this figure for clarification. The wind direction sector considered for N-6 cluster wake analysis spanning from 230° to 270° is indicated by dashed lines. (b) A/HS wind farm in the N-8 cluster with highlighted turbines for analysis: front-row turbines to the southwest (blue diamonds), reference turbines in free inflow (green diamonds) and turbines in inner-farm wake conditions (red diamonds). The origin of the coordinate system is the front-row turbine in the northwestern corner. The span-wise distance along the front row of turbines is also indicated, starting from the north east corner at [0,0] until the turbine at the southern corner, 15.8 km away from it.

**Correction of Figure 13 in the manuscript**

The labels for the axes were mistakenly switched, between unstable and stable stratification, the correction has been made. The paragraph describing the figure and sentences have also been corrected accordingly. The main difference is that the uncertainty (rather than the load effects) is higher in stable stratification, and this is attributed to the low number of cases within each stability class. There were also four cases which were wrongly labelled neutral, which has also been rectified. This causes the bars for neutral stratification to be very slightly different in magnitude from the earlier image. We apologize for the error in the pre-print. We would like to once again thank both the reviewers for bringing up the analysis of stability which enabled us to correct this error.

[Figure]

**Figure 13.** Bar graphs showing the performance indicators averaged over all turbines in a set for different atmospheric stability regimes, (a) $\zeta_{\mathrm{free}}$ and (b) $\zeta_{\mathrm{inner}}$.

**References**

A Anantharaman, G Centurelli, J Schneemann, E Bot, and M Kühn. Comparison of near wind farm wake measurements from scanning lidar with engineering models. In *Journal of Physics: Conference Series*, volume 2265, page 022034. IOP Publishing, 2022.

Marcos Paulo Araújo da Silva, Francesc Rocadenbosch, Joan Farré-Guarné, Andreu Salcedo-Bosch, Daniel González-Marco, and Alfredo Peña. Assessing obukhov length and friction velocity from floating lidar observations: A data screening and sensitivity computation approach. *Remote Sensing*, 14(6):1394, 2022.

Nicolai Cosack. *Fatigue Load Monitoring with Standard Wind Turbine Signals*. PhD thesis, University of Stuttgart, 2010.

RR Damiani. Design of offshore wind turbine towers. In *Offshore Wind Farms*, pages 263–357. Elsevier, 2016.

Francisco de N Santos, Pietro D'Antuono, Koen Robbelein, Nymfa Noppe, Wout Weijtjens, and Christof Devriendt. Long-term fatigue estimation on offshore wind turbines interface loads through loss function physics-guided learning of neural networks. *Renewable Energy*, 205:461–474, 2023.

Francisco de N Santos, Nymfa Noppe, Wout Weijtjens, and Christof Devriendt. Farm-wide interface fatigue loads estimation: A data-driven approach based on accelerometers. *Wind Energy*, 27(4): 321–340, 2024.

Martin Dörenkämper, Bjarke T Olsen, Björn Witha, Andrea N Hahmann, Neil N Davis, Jordi Barcons, Yasemin Ezber, Elena García-Bustamante, J Fidel González-Rouco, Jorge Navarro, et al. The making of the new european wind atlas–part 2: Production and evaluation. *Geoscientific Model Development Discussions*, 2020:1–37, 2020.

IEC 61400-1. International standard iec 61400-1:2019, wind energy generation systems - part 1: Design requirements. Technical report, International Electrotechnical Commission Geneva, Switzerland, 2019.

Niko Mittelmeier, Julian Allin, Tomas Blodau, Davide Trabucchi, Gerald Steinfeld, Andreas Rott, and Martin Kühn. An analysis of offshore wind farm scada measurements to identify key parameters influencing the magnitude of wake effects. *Wind Energy Science*, 2(2):477–490, 2017. doi: 10.5194/ wes-2-477-2017.

Nicolai Gayle Nygaard, Søren Trads Steen, Lina Poulsen, and Jesper Grønnegaard Pedersen. Modelling cluster wakes and wind farm blockage. In *Journal of Physics: Conference Series*, volume 1618, page 062072. IOP Publishing, 2020.

Pablo Ouro, Mina Ghobrial, Karim Ali, and Tim Stallard. Numerical modelling of offshore wind-farm cluster wakes. *Renewable and Sustainable Energy Reviews*, 215:115526, 2025.

Vasilis Pettas, Matthias Kretschmer, Andrew Clifton, and Po Wen Cheng. On the effects of inter-farm interactions at the offshore wind farm alpha ventus. *Wind Energy Science*, 6(6):1455–1472, 2021.

Clara M St Martin, Julie K Lundquist, Andrew Clifton, Gregory S Poulos, and Scott J Schreck. Atmospheric turbulence affects wind turbine nacelle transfer functions. *Wind Energy Science*, 2(1): 295–306, 2017.

Dennis D Wackerly. *Mathematical statistics with applications*. Thomson Brooks/Cole, 2008.

Qiang Wang, Kun Luo, Chunlei Wu, Junyao Tan, Rongyu He, Shitong Ye, and Jianren Fan. Inter-farm cluster interaction of the operational and planned offshore wind power base. *Journal of Cleaner Production*, 396:136529, 2023.

---

## Author Response (AR2)

**Author's response - The impact of far-reaching offshore cluster wakes on wind turbine fatigue loads**

Arjun Anantharaman, Jörge Schneemann, Frauke Theuer, Laurent Beaudet,
Valentin Bernard, Paul Deglaire, and Martin Kühn

We would like to thank both the reviewers, the associate editor and the chief editor for accepting the manuscript as is. We have addressed the technical corrections below.

**Technical corrections**

- The font size has been increased in all the figures for better readability.
- The reference was updated since their pre-print [Muller et al., 2023] was accepted and published in Wind Energy Science [Muller et al., 2024]. Only the published version is now included in our manuscript.

**References**

Etienne Muller, Simone Gremmo, Félix Houtin-Mongrolle, Bastien Duboc, and Pierre Bénard. Field data based validation of an aero-servo-elastic solver for high-fidelity les of industrial wind turbines. *Wind Energy Science Discussions*, 2023:1–38, 2023. doi: 10.5194/wes-9-25-2024.

Etienne Muller, Simone Gremmo, Félix Houtin-Mongrolle, Bastien Duboc, and Pierre Bénard. Field-data-based validation of an aero-servo-elastic solver for high-fidelity large-eddy simulations of industrial wind turbines. *Wind Energy Science*, 9:25–48, 2024. doi: 10.5194/wes-9-25-2024.